# Regulatory roles of *Escherichia coli* 5′ UTR and ORF-internal RNAs detected by 3′ end mapping

**Philip P Adams[1,2], Gabriele Baniulyte[3], Caroline Esnault[4], Kavya Chegireddy[5], Navjot Singh[3], Molly Monge[3], Ryan K Dale[4], Gisela Storz[1]\*, Joseph T Wade[3,5]\***

[1]Division of Molecular and Cellular Biology, Eunice Kennedy Shriver National Institute of Child Health and Human Development, Bethesda, United States; [2]Postdoctoral Research Associate Program, National Institute of General Medical Sciences, National Institutes of Health, Bethesda, United States; [3]Wadsworth Center, New York State Department of Health, Albany, United States; [4]Bioinformatics and Scientific Programming Core, Eunice Kennedy Shriver National Institute of Child Health and Human Development, Bethesda, United States; [5]Department of Biomedical Sciences, School of Public Health, University at Albany, Albany, United States

**Abstract** Many bacterial genes are regulated by RNA elements in their 5′ untranslated regions (UTRs). However, the full complement of these elements is not known even in the model bacterium *Escherichia coli*. Using complementary RNA-sequencing approaches, we detected large numbers of 3′ ends in 5′ UTRs and open reading frames (ORFs), suggesting extensive regulation by premature transcription termination. We documented regulation for multiple transcripts, including spermidine induction involving Rho and translation of an upstream ORF for an mRNA encoding a spermidine efflux pump. In addition to discovering novel sites of regulation, we detected short, stable RNA fragments derived from 5′ UTRs and sequences internal to ORFs. Characterization of three of these transcripts, including an RNA internal to an essential cell division gene, revealed that they have independent functions as sRNA sponges. Thus, these data uncover an abundance of *cis*- and *trans*-acting RNA regulators in bacterial 5′ UTRs and internal to ORFs.

**\*For correspondence:**
storzg@mail.nih.gov (GS);
joseph.wade@health.ny.gov (JTW)

## Introduction

The expression of many bacterial genes is controlled by elements in the 5′ untranslated regions (UTRs) of mRNAs. Changes in the secondary structures of these *cis*-acting RNA elements lead to altered expression of the associated gene(s) by modulating accessibility of ribosomes to sites of translation initiation, accessibility of RNases, or premature transcription termination. The RNA secondary structure changes can occur in response to temperature (RNA thermometers), translation of small upstream open reading frames (uORFs), or the binding of *trans*-acting factors such as metabolites (riboswitches), tRNAs, RNA-binding proteins such as CsrA, or small base-pairing RNAs (sRNAs) (reviewed in *Breaker, 2018*; *Kreuzer and Henkin, 2018*; *Loh et al., 2018*; *Orr et al., 2020*; *Romeo and Babitzke, 2019*; *Storz et al., 2011*).

Some of the regulatory events in 5′ UTRs are associated with premature transcription termination, which occurs by one of two mechanisms: intrinsic (Rho-independent) or Rho-dependent (reviewed in *Roberts, 2019*). Intrinsic termination requires only RNA polymerase and an RNA hairpin followed by a U-rich tract in the nascent RNA. Rho-dependent termination requires the loading of the hexameric Rho protein complex onto nascent, untranslated RNA at Rho utilization (Rut) sites that are typically

**eLife digest** In most organisms, specific segments of a cell's genetic information are copied to form single-stranded molecules of various sizes and purposes. Each of these RNA molecules, as they are known, is constructed as a chain that starts at the 5′ end and terminates at the 3′ end.

Certain RNAs carry the information present in a gene, which provides the instructions that a cell needs to build proteins. Some, however, are 'non-coding' and instead act to fine-tune the activity of other RNAs. These regulatory RNAs can be separate from the RNAs they control, or they can be embedded in the very sequences they regulate; new evidence also shows that certain regulatory RNAs can act in both ways.

Many regulatory RNAs are yet to be catalogued, even in simple, well-studied species such as the bacterium *Escherichia coli.* Here, Adams et al. aimed to better characterize the regulatory RNAs present in *E. coli* by mapping out the 3′ ends of every RNA molecule in the bacterium.

This revealed many new regulatory RNAs and offered insights into where these sequences are located. For instance, the results show that several of these RNAs were embedded within RNA produced from larger genes. Some were nested in coding RNAs, and were parts of a longer RNA sequence that is adjacent to the protein coding segment. Others, however, were present within the instructions that code for a protein.

The work by Adams et al. reveals that regulatory RNAs can be located in unexpected places, and provides a method for identifying them. This can be applied to other types of bacteria, in particular in species with few known RNA regulators.

C-rich, G-poor, and unstructured sequences (reviewed in *Mitra et al., 2017*). Rho translocates along the RNA until the protein catches RNA polymerase and promotes transcription termination, typically between 100 and 200 nt downstream of the Rut site, leading to 3′ ends that are processed by 3′ to 5′exonucleases (*Dar and Sorek, 2018b*; *Wang et al., 2019*).

Several studies have sought to identify sites of transcription termination across the *E. coli* genome by sequencing RNA 3′ ends or by mapping the distribution of transcribing RNA polymerase (*Dar and Sorek, 2018b*; *Ju et al., 2019*; *Peters et al., 2012*; *Peters et al., 2009*; *Yan et al., 2018*). The vast majority of identified termination sites are in 3′ UTRs. In some studies, termination was compared in cells grown with/without the Rho inhibitor, bicyclomycin (BCM), facilitating the identification of Rho termination sites (*Dar and Sorek, 2018b*; *Ju et al., 2019*; *Peters et al., 2012*). These data provided evidence for Rho termination of mRNAs in 3′ UTRs and spurious transcripts initiated within genes. While termination within 5′ UTRs was noted in some of these studies, extensive global characterization of premature termination has not been performed. Many of the uncharacterized 3′ ends in 5′ UTRs or open reading frame (ORF)-internal regions are likely to be the result of regulatory events, given that riboswitches (*Bastet et al., 2017*; *Hollands et al., 2012*), attenuators (*Ben-Zvi et al., 2019*; *Gall et al., 2016*; *Herrero Del Valle et al., 2020*; *Konan and Yanofsky, 1997*; *Kriner and Groisman, 2015*), RNA-binding proteins (*Baniulyte et al., 2017*; *Figueroa-Bossi et al., 2014*), and sRNAs (*Bossi et al., 2012*; *Sedlyarova et al., 2016*) have all been implicated in affecting premature Rho termination events.

In addition to being the product of a regulatory event, RNA fragments generated by premature termination or RNase cleavage themselves can have functions as regulatory sRNAs. sRNAs commonly base pair with *trans*-encoded mRNAs, frequently with the assistance of the RNA chaperone protein Hfq, resulting in changes in the stability or translation of the target mRNA (reviewed in *Hör et al., 2020*). Most sRNAs characterized to date are transcribed independent of other genes or are processed from mRNA 3′ UTRs, though a few 5′ UTR-derived sRNAs have been reported (reviewed in *Adams and Storz, 2020*). RNA fragments entirely internal to coding sequences (*Dar and Sorek, 2018a*) also have been suggested to function as regulators, though this has not been tested. While sRNAs generally base pair with mRNA targets, a few small transcripts have been shown to have roles as competing endogenous RNAs (ceRNAs) also known as 'sponges', which base pair primarily with sRNAs, targeting the sRNAs for degradation or blocking their interactions with mRNA targets (reviewed in *Denham, 2020*; *Figueroa-Bossi and Bossi, 2018*; *Grüll and Massé, 2019*).

To systematically identify new regulatory elements in *E. coli*, we globally mapped RNA 3′ ends, and specifically characterized those ends in 5′ UTRs and ORF-internal regions. We compared this 3′ end dataset with another dataset where BCM treatment was used to identify sites of Rho termination. Using these approaches, we detected hundreds of RNA 3′ ends within 5′ UTRs and internal to ORFs, likely generated by premature transcription termination or RNase processing. We propose the majority of these 3′ ends are the consequence of regulatory events, and we document regulation for multiple examples. For instance, we show 3′ ends are associated with the translation of uORFs, or result from the binding of some sRNAs to mRNA 5′ UTRs. Furthermore, we demonstrate that RNA fragments generated by premature transcription termination and from within coding sequences function as independent sRNA regulators; one as part of an autoregulatory loop and another that connects cell division to the cell envelope stress response. These findings reveal extensive and diverse regulation through premature transcription termination and RNase processing of mRNAs, which can lead to the generation of RNA by-products with independent functions.

## Results

### Global mapping of 3′ ends in *E. coli*

Two independent cultures of wild-type *E. coli* MG1655 (WT) were grown to $OD_{600}$ ~0.4 in rich (LB) medium, $OD_{600}$ ~2.0 in LB, and $OD_{600}$ ~0.4 in minimal (M63) glucose medium. Total RNA was isolated and analyzed using modified RNAtag-seq (*Shishkin et al., 2015*) (total RNA-seq) and 3′ end (Term-seq) protocols (*Dar et al., 2016*; *Figure 1—figure supplement 1*). The replicate total RNA-seq and Term-seq datasets were highly correlated (*Figure 1—figure supplement 2A*). Using the Term-seq data, we curated a list of dominant RNA 3′ ends (*Supplementary file 1*). The total numbers of identified 3′ ends were 1175 and 882 for cells grown in LB to $OD_{600}$ ~0.4 or 2.0, respectively, and 1053 for cells grown in M63 glucose to $OD_{600}$ ~0.4 (*Figure 1—figure supplement 2B*). The detected 3′ ends were further subclassified (see Materials and methods for details) according to their locations relative to annotated genes (*Figure 1A*). 3′ ends that could not be assigned to one unique category were counted in multiple categories. For cells grown to $OD_{600}$ ~0.4 in LB, this analysis revealed that, while 23% of 3′ ends mapped <50 bp downstream of an annotated gene (primary 3′ ends), hundreds (58%) were classified as orphan and internal 3′ ends, many mapping upstream of, and within ORFs (*Figure 1B*).

We compared the detected 3′ ends to those identified by three other RNA-seq based studies (*Dar and Sorek, 2018b*; *Ju et al., 2019*; *Yan et al., 2018*). Note that the number of overlapping 3′ ends differs depending on the direction of the comparison (is non-commutative) because multiple 3′ ends from one dataset can be close to a single 3′ end from the other dataset. While there was significant overlap between the 3′ ends identified in our work and those identified in each of the other studies (*Figure 1—figure supplement 2C,E,F*; hypergeometric test p<2.2e$^{-16}$ in all three cases), the majority were unique to our study. Given that a previous *E. coli* Term-seq study only reported 3′ ends downstream of annotated genes (*Dar and Sorek, 2018b*), we used our 3′ end calling algorithm to re-analyze the sequencing data from this study (*Figure 1—figure supplement 2D*). Again, there was significant overlap (hypergeometric test p<2.2e$^{-16}$), but 51% were unique to our study. Here, we focused on, and characterized 3′ ends mapping upstream of, and internal to, coding sequences, most of which were not detected in the other RNA-seq studies.

### Global mapping of Rho termination regions in *E. coli*

Concurrently, we mapped instances of Rho-dependent transcription termination. One *E. coli* MG1655 culture grown to $OD_{600}$ ~0.5 in LB was split with half left untreated and the other half treated with 100 µg/ml BCM for 15 min. Total RNA was isolated and analyzed using a DirectRNA-seq protocol (*Ozsolak and Milos, 2011*; *Figure 1—figure supplement 1*), a method optimized for sequencing very short reads (20–30 nt) directly from RNA, an advantage for examining the effect of Rho on the generation of small RNA fragments. Using these data, we calculated a 'Rho score' for each genomic position by comparing DirectRNA-seq coverage in 800 nt windows upstream and downstream in the treated (+BCM) and untreated (-BCM) samples (see Materials and methods). This ratio reflects the degree of transcriptional Rho readthrough in the +BCM cells, where a score of >2.0 is indicative of at least a two-fold increase in readthrough in the +BCM compared to the -

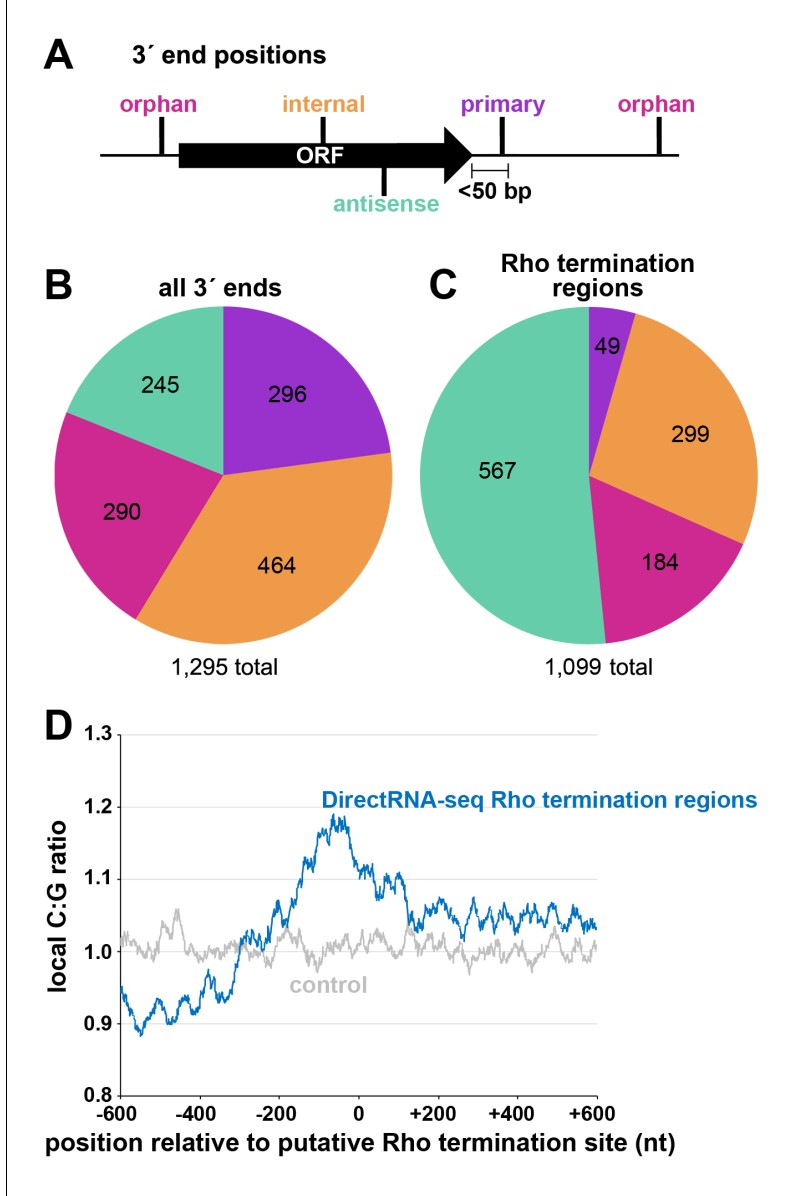

**Figure 1.** Distribution of 3′ ends and putative sites of Rho termination. (**A**) Schematic of classification of Term-seq 3′ ends and Rho termination sites relative to an annotated ORF. 3′ ends and termination sites were defined as: primary (purple colored, located on the same strand within 50 bp downstream of the 3′ end of an annotated gene (mRNA ORF, tRNA, rRNA, or sRNA)), antisense (aquamarine colored, located on the opposite strand within 50 bp of a gene start and end coordinates), internal (orange colored, located on the same strand within a gene) and orphan (fuchsia colored, located in a 5′ UTR, long 3′ UTR or not falling in any of the previous classes). The black arrow represents an ORF. (**B**) Distribution of Term-seq 3′ ends relative to annotated genes. Some 3′ ends fit the criteria for two different categories; 99 are primary and antisense, eight are primary and internal and 13 are internal and antisense. (**C**) Distribution of Rho termination sites relative to annotated genes based on DirectRNA-seq after BCM treatment. Some fit the criteria for two different categories; 12 are primary and antisense and six are primary and internal, three are antisense and internal. (**D**) C:G ratio of sequences surrounding predicted Rho termination sites. Nucleotide proportions were calculated by scanning 600 nt upstream and downstream of Rho 3′ ends (*Supplementary file 2*) using 25 nt windows. Plotted values represent the average ratios for all 1078 regions (blue). Control plot (gray) represents average C:G ratios calculated in the same manner for 1078 random *E. coli* MG1655 genomic positions.

The online version of this article includes the following figure supplement(s) for figure 1:

**Figure supplement 1.** RNA-seq approaches.

**Figure supplement 2.** Analysis of Term-seq and DirectRNA-seq data.

BCM sample. 1078 genomic regions were putatively associated with a Rho termination event (*Supplementary file 2*). We note that these genomic positions are likely closer to processed RNA 3′ ends than the termination sites, since Rho-terminated transcripts typically are processed by 3′ to 5′ exonucleases (*Dar and Sorek, 2018b*; *Wang et al., 2019*). Hence, we refer to the identified genomic locations as 'Rho termination regions'. As for 3′ ends mapped by Term-seq, we classified Rho termination regions by their position relative to annotated genes (*Figure 1C*). 3′ ends antisense to annotated genes represented the largest category (51.5%), consistent with significant Rho termination of antisense RNAs (*Peters et al., 2012*). Only a small percentage (4.5%) of the Rho termination regions are <50 nt downstream of an annotated gene (primary), likely because Rho loading and termination typically requires >50 nt untranslated RNA (reviewed in *Mitra et al., 2017*). As for the 3′ end-mapping by Term-seq, many Rho termination regions (44%) were classified as orphan and internal, frequently mapping upstream of, or within ORF sequences.

The C:G nucleotide usage was calculated for the putative Rho termination regions (*Supplementary file 2*), as well as for a control group of randomly selected genomic coordinates. Relative to the control group, there was a higher local C:G ratio within 200 nt of the 3′ ends associated with Rho termination regions (*Figure 1D*), consistent with enrichment for C-rich, G-poor sequences attributed to Rut sites (reviewed in *Mitra et al., 2017*).

We also compared the Rho termination regions (*Supplementary file 2*) to the sites of Rho termination reported in three previous genome-wide studies (*Dar and Sorek, 2018b*; *Ju et al., 2019*; *Peters et al., 2012*; *Figure 1—figure supplement 2G*). Again, there was significant overlap with each of the three previous studies (hypergeometric test $p<2.2e^{-16}$ in all cases), though many (43%) of the putative Rho termination regions we identified are >500 bp away from any of the previously identified Rho termination regions (*Dar and Sorek, 2018b*; *Ju et al., 2019*; *Peters et al., 2012*). The large sets of Rho termination regions that differ between the studies suggest that much remains to be learned about Rho-dependent termination. The comparison of our Term-seq LB 0.4 dataset (*Supplementary file 1*) with our DirectRNA-seq LB 0.5 dataset (*Supplementary file 2*) revealed that there was significant overlap (hypergeometric test $p<2.2e^{-16}$), with 34.6% of the Term-seq 3′ ends within 500 nt of a Rho termination region (*Figure 1—figure supplement 2H*). This suggests that these 3′ ends are associated with Rho-terminated transcripts.

## Known regulatory events are associated with 3′ ends and Rho termination regions in 5′ UTRs

Many 3′ ends identified by Term-seq mapped upstream or internal to annotated mRNAs (*Supplementary file 1*). We specifically focused on these 3′ ends (see Materials and methods), since we hypothesized they could be reflective of regulatory events. Our analysis identified 665 and 507 3′ ends for the LB 0.4 and LB 2.0 samples, respectively, and 580 3′ ends for the M63 0.4 sample (*Supplementary file 3*). Among the 3′ ends in 5′ UTRs, several correspond to sites of known *cis*-acting RNA regulation (annotated in *Supplementary file 3*). These include mRNAs that previously have been shown to be regulated by premature Rho-dependent termination, such as the riboswitch-regulated genes *thiM* (*Bastet et al., 2017*; *Figure 2—figure supplement 1A*), *mgtA* (*Hollands et al., 2012*), *ribB* (*Hollands et al., 2012*) and *lysC* (*Bastet et al., 2017*), and the translationally-repressed genes *ilvX* (*Lawther and Hatfield, 1980*) and *topAI* (*Baniulyte and Wade, 2019*). We also noticed that even some 5′ UTRs where regulation has only been reported to be at the level of translation, such as the RNA thermometer upstream of *rpoH* (*Morita et al., 1999a*; *Morita et al., 1999b*) and at ribosomal protein operons (reviewed in *Zengel and Lindahl, 1994*), harbored defined 3′ ends in 5′ UTRs.

For the LB 0.4 Term-seq dataset, for which the growth conditions were most similar to those of the DirectRNA-seq dataset, we determined whether each 3′ end was associated with a detected Rho termination event. Significant Rho scores are listed in *Supplementary file 3* (see Materials and methods for details). It should be noted that some genes containing 3′ ends with significant Rho scores in *Supplementary file 3* are absent from *Supplementary file 2*, because the genomic position with the highest Rho score in that region is located in a neighboring gene/region, and thus given a different gene designation.

We also used a previously described quantitative model to score putative intrinsic terminators, where a score >3.0 is predictive of intrinsic termination (*Chen et al., 2013*). Based on these analyses for the 3′ ends for the LB 0.4 dataset in *Supplementary file 3*, we predict that 20% have secondary

structures and sequences consistent with intrinsic termination (intrinsic terminator score ≥3.0), 11% have detectable Rho termination, and the remaining 3′ ends are likely the result of RNA processing. However, the number of Rho termination sites could be an underestimate, because inhibition of Rho leads to extensive readthrough transcription from very strongly transcribed genes that can mask overlapping transcripts, and 3′ ends generated by Rho termination are typically unstable (*Dar and Sorek, 2018b*; *Wang et al., 2019*). Some 3′ends also may be generated by a combination of mechanisms. Nonetheless, overall, our data suggest that modulation of premature transcription termination in 5′ UTRs or ORFs is a widespread regulatory mechanism.

## Novel sites of regulation are predicted by 3′ ends and Rho termination regions in 5′ UTRs

Several genes harboring Rho-dependent 3′ ends within the 5′ UTR or ORF have not been previously described as being regulated by Rho but are associated with characterized *cis*-acting RNA regulators. Examples are the *sugE* (*gdx*) and *moaA* genes, which are preceded by the guanidine II riboswitch (*Sherlock et al., 2017*) and molybdenum cofactor riboswitch (*Regulski et al., 2008*), respectively. A browser image of the RNA-seq data for the *sugE* locus in the LB 0.4 condition documents a predominant 3′ end 76 nt downstream the *sugE* transcription start site (TSS) (*Figure 2A*). This region was associated with significant readthrough in the +BCM DirectRNA-seq sample and a 3′ end Rho score of 3.7 (*Supplementary file 3*), strongly suggesting that the riboswitch impacts Rho-dependent premature termination, as is the case for other riboswitches.

While the involvement of 5′ UTRs in *sugE* and *moaA* regulation was known, most of the genes for which we found 3′ ends in 5′ UTRs or ORFs have not been reported to have RNA-mediated regulation (*Supplementary file 3*). In some cases, such as the *mdtJI* mRNA, encoding a spermidine efflux pump, we observed a novel 3′ end that is clearly Rho-dependent (*Figure 2B*) with a Rho score of 2.3. In other cases, such as *ispU* (*uppS*), encoding the undecaprenyl pyrophosphate synthase, a novel 3′ end was observed with no readthrough upon Rho inhibition (*Figure 2C*; Rho score of 0.7). Rather, this 3′ end had an intrinsic terminator score (*Chen et al., 2013*) of 14.1 (*Supplementary file 3*), and we predicted a 6 bp stem-loop followed by eight U residues, consistent with intrinsic termination.

To further test whether genes associated with 3′ ends in 5′ UTRs or ORFs are indeed regulated by premature Rho-dependent transcription termination, we generated *lacZ* transcriptional reporter fusions using the entire 5′ UTR and ORF for 27 genes (*Supplementary files 2* and *3*) arbitrarily chosen for a range of calculated Rho scores and possible regulation. All constructs had the same constitutive promoter. β-galactosidase activity was assayed in the context of a WT *E. coli* background or a mutant strain with an R66S substitution in Rho (*rhoR66S*), which disrupts the primary RNA-binding site (*Baniulyte et al., 2017*; *Bastet et al., 2017*; *Martinez et al., 1996*). A fusion to *thiM*, which harbors a 5′ UTR Rho-dependent terminator (*Bastet et al., 2017*), exhibited significantly higher levels of β-galactosidase activity in the Rho mutant strain, indicative of a disruption in Rho-dependent termination (*Figure 2—figure supplement 1B*). Northern analysis identified an RNA consistent with the 3′ end identified by Term-seq (*Figure 2—figure supplement 1C*), though we noted the abundance of this 5′ RNA fragment and extent of readthrough in the Rho R66S mutant varied with growth phase.

Among the other constructs assayed, the expression of 14 fusions was >2-fold higher in *rhoR66S* compared to WT cells (*Figure 2D* and *Figure 2—figure supplement 1D*), consistent with Rho termination in the 5′ UTR or ORF for *sugE*, *cfa*, *cyaA*, *mdtJ*, *add*, *cspB*, *cspG*, *moaA*, *pyrG*, *yhaM*, *ydjL*, *yhiI*, *ytfL*, and *yajO*. The effect of the Rho mutation on the *eptB*, *chiP*, and *crp-yhfK* fusions was intermediate (1.6- to 1.8-fold) (*Figure 2D*), while the fusions to *ispU* (*Figure 2D*), as well as *mnmG*, *rpsJ*, *argT*, *srkA*, and *trmL* (*Figure 2—figure supplement 1D*), displayed similar levels of β-galactosidase activity (1.0- to 1.4-fold) for *rhoR66S* compared to WT cells. These assays support the notion that the 3′ end observed in the *ispU* 5′ UTR (*Figure 2C*) is generated by intrinsic termination. It is unclear why we did not detect evidence for Rho termination of *mnmG*, *rpsJ*, *argT*, *srkA* or *trmL*, despite these genes having significant Rho scores. Interestingly, fusions to the *ompA*, *yebO*, *glpF*, and *rimP* 5′ UTR and ORF had decreased expression in the Rho mutant background (*Figure 2—figure supplement 1D*). This may be a consequence of additional levels of regulation or indirect effects of Rho inhibition. The *ompA* and *glpF* genes were not associated with Rho termination by DirectRNA-seq.

Transcriptional *lacZ* reporter gene fusions to only the 5′ UTR (i.e. without the ORF) were also generated for seven genes, to distinguish between Rho termination in the ORF and in the 5′ UTR. The

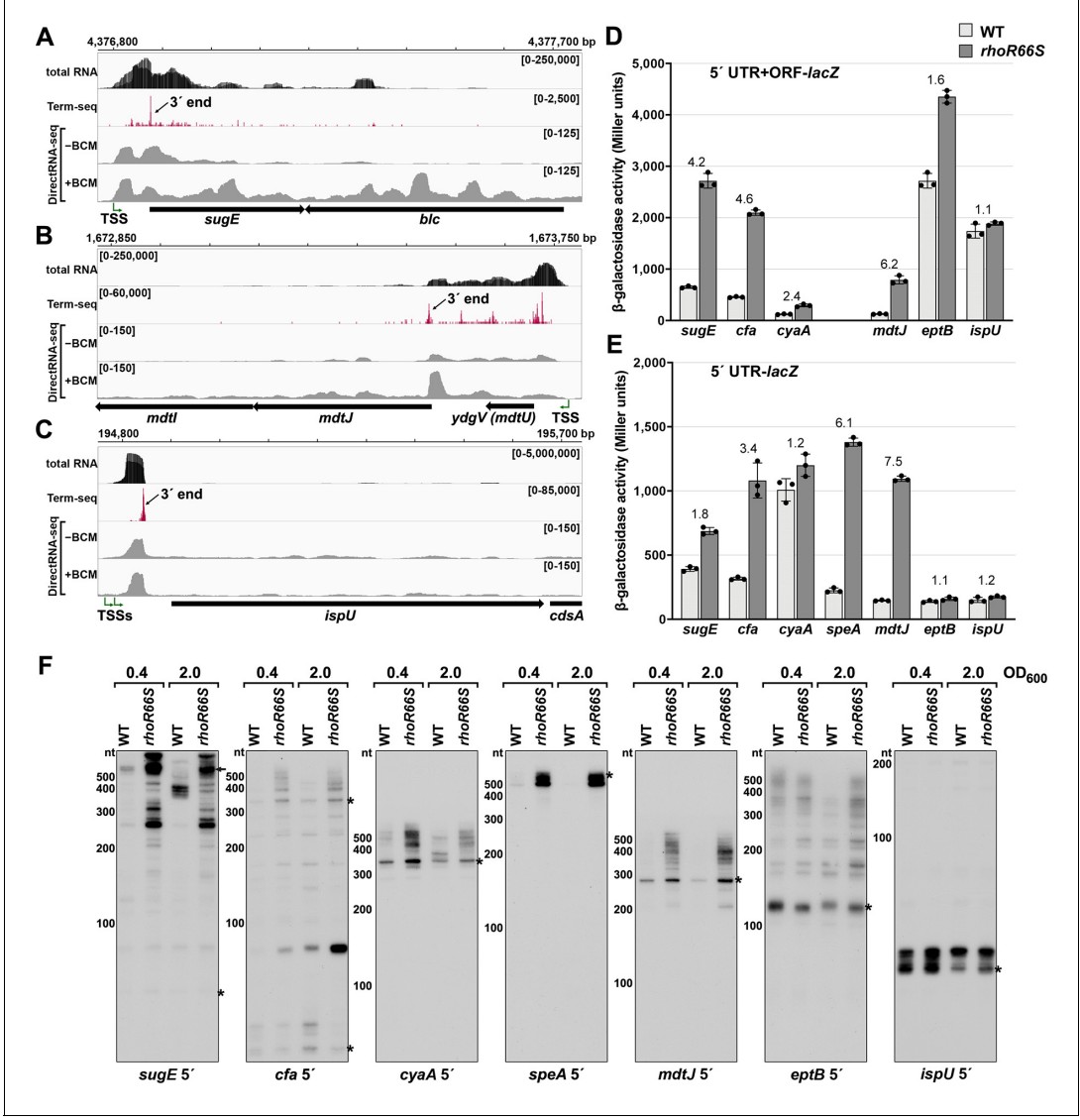

**Figure 2.** Experimental validation of premature Rho termination. (**A**) RNA-seq screenshot of the *sugE* (*gdx*) locus displaying sequencing reads from LB 0.4 total RNA-seq, LB 0.4 Term-seq and DirectRNA-seq ±BCM treatment. Total and Term-seq tracks represent an overlay of two biological replicates. Read count ranges are shown in the top of each frame. The chromosome nucleotide coordinates, relative orientation of the *sugE* and *blc* ORFs (wide black arrows), dominant 3′ end in the *sugE* 5′ UTR from **Supplementary file 3** (small black arrow labeled 3′ end), and *sugE* TSS (**Thomason et al., 2015**) (bent green arrow) are indicated. (**B**) RNA-seq screenshot of the *mdtJI* locus, labeled as in (**A**). (**C**) RNA-seq screenshot of the *ispU* (*uppS*) locus, labeled as in (**A**). (**D**) β-galactosidase activity for *sugE*, *cfa*, *cyaA*, *mdtJ*, *eptB* and *ispU* 5′ UTR + ORF transcriptional fusions to *lacZ* in WT (AMD054) and *rhoR66S* mutant (GB4). All gene-reporter fusions initiate from the same high expression promoter and were assayed at OD<sub>600</sub> ~0.4–0.6 (see Materials and methods for details). Values represent the mean of at least three independent replicates (indicated by black dots). Error bars represent one standard deviation from the mean. The *rhoR66S* vs WT fold change is reported above the values for each 5′ UTR. A *speA* 5′ UTR + ORF-*lacZ* could not be assayed because cells did not grow, likely because of toxicity associated with overexpression of the full-length gene product. (**E**) β-galactosidase activity for *sugE*, *cfa*, *cyaA*, *speA*, *mdtJ*, *eptB*, and *ispU* 5′ UTR transcriptional fusions to *lacZ* in WT (AMD054) and *rhoR66S* mutant (GB4). Experiments were performed and data analyzed as in (**D**). DirectRNA-seq Rho scores for the dominant 3′ end in the 5′ UTR (from **Supplementary file 3**) of these loci are: 3.7 for *sugE*, 2.6 for *cfa*, 2.0 and 2.6 for *cyaA* (there are two 3′ ends), 3.1 for *speA*, 2.3 for *mdtJ*, 2.3 for *eptB* and 0.7 for *ispU*. Rho termination regions were also identified in these genes, or neighboring genes, for all loci except *ispU* (**Supplementary file 2**). See **Supplementary file 4** for oligonucleotides used in cloning to delineate regions in each *lacZ* fusion. (**F**) Northern analysis for *sugE*, *cfa*, *cyaA*, *speA*, *mdtJ*, *eptB* and *ispU* 5′ UTRs in WT (GSO989) and *rhoR66S* mutant (GSO990) cells. Cells were grown to OD<sub>600</sub> ~0.4 or 2.0 after a dilution of the overnight culture and lysed. Total RNA was extracted, separated on an acrylamide gel, transferred to a membrane and probed for the indicated RNAs (RNAs were probed sequentially on the same membrane). Blot was also probed for 5S (**Figure 2—figure supplement 1C**). Size markers are indicated for all RNAs. Asterisks signify the transcript predicted to correlate to the 3′ end in **Supplementary file 3**. Arrow points to expected full-length *sugE* transcript.

The online version of this article includes the following figure supplement(s) for figure 2:

*Figure 2 continued on next page*

**Figure supplement 1.** Test of Rho-dependent termination in several genes.

effect of *rhoR66S* was eliminated for the shorter *cyaA* and *eptB* fusions (**Figure 2E**), suggesting that Rho-dependent termination occurs within the coding sequence of these genes. Regulation of Rho termination in 5′ UTRs is probably associated with the accessibility of Rut sequences, whereas Rho termination within coding sequences is probably associated with regulated translation initiation (reviewed in **Kriner et al., 2016**), with translational repression indirectly leading to Rho termination.

Finally, using RNA extracted from WT and *rhoR66S* strains grown to $OD_{600}$ ~0.4 and 2.0 in LB, northern analysis was performed with probes for the 5′ UTRs of *sugE*, *cfa*, *cyaA*, *speA*, *mdtJ*, *eptB* and *ispU* (**Figure 2F**) as well as with probes for the coding sequences of *cfa*, *cyaA*, *speA* and *mdtJ* (**Figure 2—figure supplement 1E**). For all of the mRNAs, we detected 5′ fragments that likely correspond to the 3′ ends detected by Term-seq (indicated with an asterisk) (**Supplementary file 3**). For *sugE* and *cfa*, however, the dominant band on the northern blot was not necessarily the most dominant 3′ end sequenced using Term-seq. The enrichment for the 5′ fragments as well as longer transcripts likely corresponding to the full-length *sugE*, *cfa*, *cyaA*, *speA* and *mdtJ* mRNAs (indicated with an arrow) in the *rhoR66S* samples reflect transcriptional readthrough in the mutant background. This is consistent with Rho-dependent termination as seen for the *lacZ* fusions. The effect of *rhoR66S* on *eptB* was intermediate, while no effect was observed for *ispU*. For all of the Rho-terminated genes, we were surprised to observe the significant increase in the levels of short transcripts, as seen most strikingly for *speA*. This suggests that the detected 3′ ends can be generated by RNA processing from both Rho-terminated and full-length transcripts, with the increased abundance of longer transcripts in the *rhoR66S* samples leading to higher levels of the processed product. We also noted that the effects of the *rho* mutation varied under the different growth conditions tested, as is most apparent for *cfa* and *cyaA*. Collectively, these data validate premature termination in 5′ UTRs and, in several cases, suggest complex regulation.

## Premature Rho termination of *mdtJI* is dependent on spermidine and translation of a uORF *mdtU*

The *mdtJI* mRNA, encoding a spermidine exporter, has a long 5′ UTR (TSS located 278 nt from the start codon) (**Figure 3A**) and a 3′ end that mapped six nt into the ORF (**Figure 2B**). Additionally, the transcript is subject to premature Rho termination (**Figure 2D–F**, **Figure 2—figure supplement 1E**). The levels of the *mdtJI* transcript were previously reported to increase in response to high levels of spermidine, a polyamine (**Higashi et al., 2008a**), though a mechanism for this regulation was not described. Polyamines play important roles in RNA-mediated regulation, reported to cause structural changes to the 5′ UTR of the *oppA* mRNA (**Higashi et al., 2008b**; **Yoshida et al., 1999**) and induce ribosome stalling at a uORF in the 5′ UTR of the *speFL* mRNA (**Ben-Zvi et al., 2019**; **Herrero Del Valle et al., 2020**). In light of these studies, we examined the effect of spermidine on *mdtJI* mRNA levels. Total RNA was extracted from cells grown in LB medium with or without 10 mM spermidine at either neutral or high pH (spermidine and other polyamines have a stronger negative effect on cell growth at high pH **Yohannes et al., 2005**). Northern analysis of these samples probed for the *mdtJI* mRNA revealed a ~280 nt transcript, consistent with the 3′ end detected by Term-seq (**Supplementary file 3**), that was susceptible to readthrough upon the addition of spermidine for cells grown at high pH (**Figure 3B**). We therefore hypothesized that spermidine inhibits premature Rho termination of the *mdtJI* mRNA.

Closer inspection of the *mdtJI* 5′ UTR revealed a putative upstream small ORF (uORF) of 34 codons with the stop codon 106 nt upstream of the *mdtJ* AUG (**Figure 3A**). Translation of this uORF was previously detected by ribosome profiling, with expression of the corresponding small protein (YdgV) documented by western blot analysis (**Weaver et al., 2019**). To independently verify translation of the uORF, the coding sequence was translationally fused to *lacZ* together with the upstream sequence and native *mdtUJI* promoter. Robust β-galactosidase activity was detected for cells carrying the fusion (**Figure 3C**). By contrast, no β-galactosidase activity was observed for cells carrying an equivalent fusion with the uORF start codon mutated (ATG→ACG), supporting the conclusion that the uORF (*ydgV*), here renamed *mdtU*, is indeed translated.

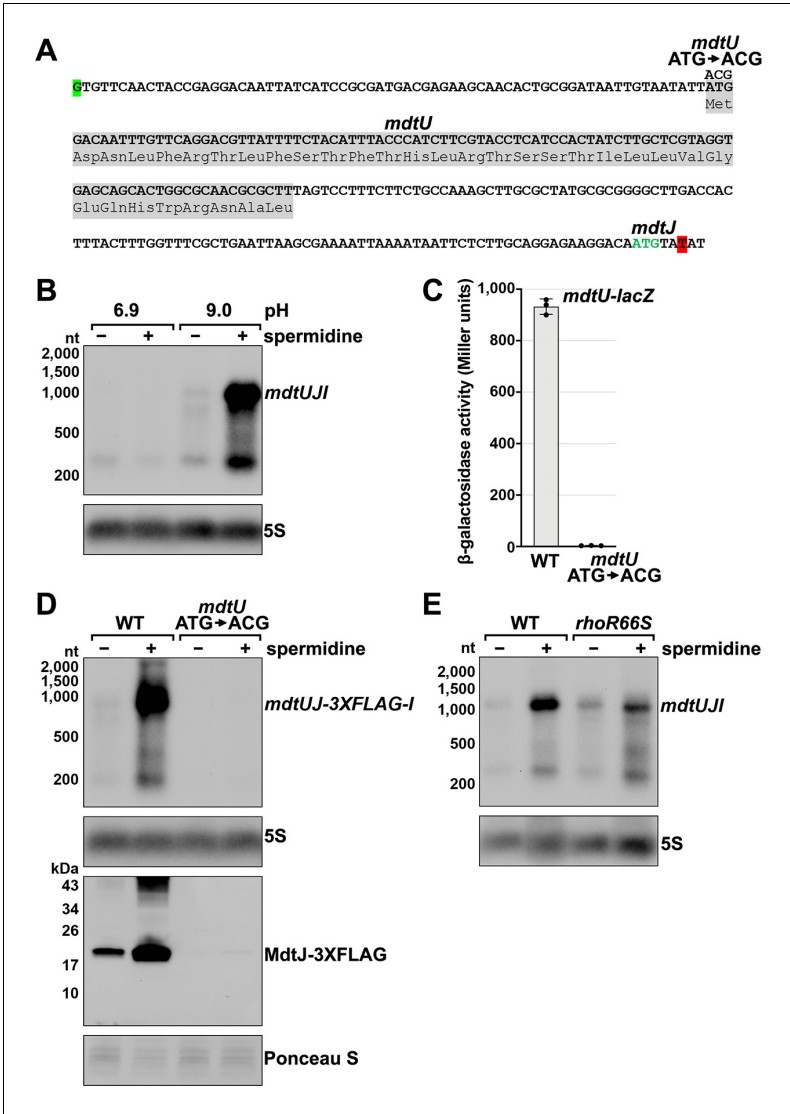

**Figure 3.** Effect of spermidine on *mdtUJI* expression. (**A**) Sequence of the *mdtJI* 5′ UTR. The transcription start site (green shaded nucleotide) determined by dRNA-seq (*Thomason et al., 2015*) and 3′ end (red shaded nucleotide) determined by Term-seq (current study) are indicated. Sequence encoding the *mdtU* uORF is highlighted in gray. Start codon of the *mdtJ* ORF is indicated with green text. (**B**) Northern analysis of effects of spermidine on *mdtUJI* mRNA levels. WT (GSO989) cells were grown for 150 min after a dilution of the overnight culture, ±10 mM spermidine in either LB pH 6.9 or LB pH 9.0. Total RNA was extracted, separated on an agarose gel, transferred to a membrane and sequentially probed for the *mdtJI* 5′ UTR and 5S. (**C**) β-galactosidase activity of a *mdtU* translational *lacZ* fusion. WT *mdtU* (pASW1) and start codon (ATG→ACG) mutant (pGB337) were assayed and analyzed as in *Figure 2D*. Constructs included the native *mdtUJI* TSS and full-length *mdtU* ORF. (**D**) Northern and western analyses of the effect of the *mdtU* uORF mutant on *mdtJ-3XFLAG-mdtI* mRNA and MdtJ-3XFLAG levels. WT *mdtU* (GSO991) and start codon (ATG→ACG) mutant (GSO992) cells harboring an *mdtUJ-3XFLAG-I* were grown for 150 min after a dilution of the overnight culture, ±10 mM spermidine in LB pH 9.0. Total RNA was analyzed as in (**B**). Protein extracts were separated on a Tris-Glycine gel, transferred to a membrane, stained using Ponceau S stain, and probed using α-FLAG antibodies. We do not know the identity of the higher molecular weight bands observed for the WT sample in the western analysis. They could be due to multimeric MdtJ or MdtJ association with the membrane. (**E**) Northern analysis of Rho effect on *mdtUJI* mRNA levels in the presence of spermidine. WT (GSO989) and *rhoR66S* mutant (GSO990) cells were grown for 150 min after a dilution of the overnight culture, ±10 mM spermidine in LB pH 9.0. Total RNA was analyzed as in (**B**).

The online version of this article includes the following figure supplement(s) for figure 3:

**Figure supplement 1.** Amino acid conservation of *mdtU* uORF in selected gammaproteobacterial species.

To investigate the role of *mdtU* in spermidine-mediated regulation of *mdtUJI*, the *mdtU* start codon mutation (ATG→ACG) was introduced on the chromosome of a strain where a 3XFLAG tag was translationally fused to C-terminus of MdtJ. Northern and western blot analysis of strains encoding *mdtUJ-3XFLAG-I* showed mRNA and protein levels were strongly induced by spermidine, which was abolished in the strain with the *mdtU* start codon mutation (*Figure 3D*). We suggest that translation of MdtU is critical for spermidine-mediated expression of MdtJ. Northern analysis was also carried out to determine if Rho termination in the *mdtJI* 5′ UTR impacts the induction by spermidine. In the *rhoR66S* strain, the spermidine-dependent increase in full-length *mdtUJI* mRNA levels was substantially reduced relative to WT cells (*Figure 3E*). However, inhibition of Rho did not completely abolish the stimulatory effect of spermidine on *mdtUJI* mRNA levels, perhaps because growth in spermidine and high pH may increase transcription initiation of *mdtUJI*. Together, these data support the model that spermidine, Rho, and translation of the *mdtU* uORF affect the levels of MdtJI and hence spermidine transport, though the mechanisms deserve further study. A screen for uORFs upstream of *mdtJ* orthologs in other gammaproteobacterial species showed that MdtU, particularly the C-terminal region, is conserved in at least 17 genera (*Figure 3—figure supplement 1*). This suggests that *mdtU*-dependent regulation of *mdtUJI* expression is a conserved process that may depend on the sequence of the MdtU C-terminus.

The presence of 3′ ends downstream of putative uORFs could be a way to identify new uORF-dependent regulatory sequences. A search for 3′ ends located <200 nt downstream of experimentally validated, but uncharacterized potential uORFs (*Hemm et al., 2008*; *VanOrsdel et al., 2018*; *Weaver et al., 2019*) revealed nine examples: *ybgV-gltA*, *yhiY-yhiI*, *ykiE-insA-7*, *yliM-ompX*, *ymdG-putP*, *ymiC-acnA*, *yqgB-speA*, and *ytgA-iptF* (*Supplementary file 3*), consistent with this suggestion.

## 5′ mRNA fragments can be generated by sRNA-mediated regulation

The vast majority of characterized regulatory binding sites for sRNAs are in 5′ UTRs, and we observed several RNA 3′ ends in 5′ UTRs near positions of documented sRNA base pairing (*Supplementary file 3*). For instance, the *eptB*, *ompA* and *chiP* mRNAs, which are targets of MgrR (*Moon and Gottesman, 2009*), MicA (*Udekwu et al., 2005*) and ChiX (*Figueroa-Bossi et al., 2009*), respectively, all had 3′ ends directly downstream of the sequences involved in sRNA base pairing (*Figure 4A* and *Figure 4—figure supplement 1*). The presence of RNA 3′ ends at these positions suggested that stable 5′ mRNA fragments could be generated or perhaps protected by sRNA-mediated regulation. Indeed, 5′ transcripts for *eptB* and *chiP* were previously detected by total RNA-seq (*Dar and Sorek, 2018a*), and the *chiP* 5′ transcript was reported to accumulate in the absence of the 3′-to-5′ phosphorolytic exoribonuclease PNPase (*Cameron et al., 2019*).

To examine how sRNAs impacted the 5′ derived mRNA fragments of *eptB*, *ompA,* and *chiP*, we used northern analysis to examine the consequences of deleting or overexpressing the cognate sRNA gene. Two oligonucleotide probes, one within the coding sequence (downstream of the 3′ end identified using Term-seq) and one within the 5′ fragment, were used to determine the relative levels of the full-length mRNAs (*Figure 4B*) and 5′ fragments (*Figure 4C*). As expected, given the known sRNA-mediated downregulation of *eptB*, *ompA* and *chiP*, the levels of the target mRNAs were elevated in the sRNA deletion background compared to the WT strain and decreased with sRNA overexpression (*Figure 4B*). Some other RNA species were detected for *eptB* (~400 nt), *ompA* (<3000 nt), and *chiP* (~200–500 nt), but these did not match the expected sizes for the mRNAs, and may be degradation and/or readthrough products. For the 5′ UTR region of *eptB*, there was a reciprocal effect of the Δ*mgrR* background, with a strong decrease in the abundance of a ~140 nt band (*Figure 4C*). Given that we observed a moderate effect of Rho mutation on *eptB* expression (*Figure 2D,F*), we speculate that MgrR base pairing both promotes Rho termination and protects the resultant RNA from exonucleases. For the 5′ UTR region of *ompA*, which showed only modest de-repression in the Δ*micA* strain, there was a decrease in the abundance of an ~120 nt fragment in the deletion strain (*Figure 4C*), and the levels of this fragment increased upon MicA overexpression. These effects are consistent with MicA sRNA-directed cleavage of the *ompA* mRNA generating the fragment, or the sRNA base pairing protecting the 3′ end from exonucleolytic processing. The effect of the Δ*chiX* background on the *chiP* 5′ fragment was strikingly different. Instead of decreasing, there was a large increase in the levels of an ~90 nt RNA (*Figure 4C*). Given the strong signal detected for this transcript, we hypothesized that this RNA might have an independent role as a regulatory RNA.

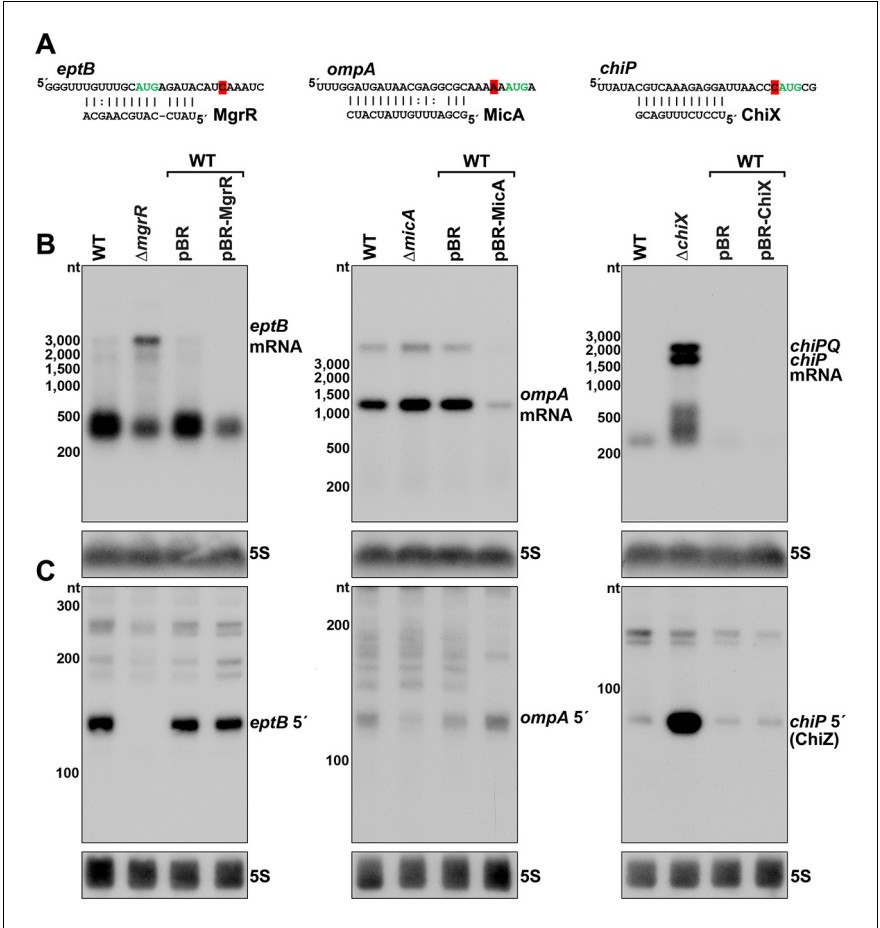

**Figure 4.** Effect of sRNA deletions on *eptB*, *ompA,* and *chiP* fragments. (**A**) Sequence of documented region of sRNA-mRNA pairing. 3′ end determined by Term-seq is highlighted in red. Start codon of the corresponding ORF is indicated with green text. (**B**) Northern analysis of *eptB*, *ompA*, and *chiPQ* mRNAs. WT (GSO982) without and with indicated plasmids and Δ*mgrR* (GSO993), Δ*micA* (GSO994), and Δ*chiX* (GSO995) cells were grown for 150 min after a dilution of the overnight culture. Total RNA was extracted, separated on an agarose gel, transferred to a membrane and sequentially probed for specific mRNAs and 5S. Size markers are indicated for all RNAs. (**C**) Northern analysis of *eptB*, *ompA*, and *chiP* 5′ UTR fragments. The same RNA as in (**B**) was separated on an acrylamide gel, transferred to a membrane and probed for specific 5′ UTR fragments and 5S. Size markers are indicated for all RNAs.

The online version of this article includes the following figure supplement(s) for figure 4:

**Figure supplement 1.** Sequences of *eptB*, *ompA* and *chiP* 5′ UTRs.

## ChiZ and IspZ sRNA sponges derive from 5′ UTRs

To test the hypothesis that 5′ UTR transcripts with defined bands have independent functions as sRNAs, we carried out further studies on the ~90 nt *chiP* 5′ UTR transcript (*Figure 4C*), which we renamed ChiZ, and the 81 and 60 nt *ispU* 5′ UTR transcripts (likely expressed from two TSSs with a shared 3′ end, *Figure 2C and F*), denoted IspZ (*Figure 5A*).

To obtain more information about the expression of these putative sRNAs, we performed northern analysis using the same RNA analyzed in the Term-seq experiment (*Figure 5B*). Distinct bands were detected for the two 5′ UTRs, consistent with the generation of stable RNAs with predicted stems protecting the ends (*Figure 5—figure supplement 1A–B*). While ChiZ was most abundant in cells grown to exponential phase in LB, IspZ was abundant in LB and M63 glucose medium in both exponential and stationary phase. Since many sRNA levels are negatively affected by the lack of the RNA chaperone Hfq (reviewed in *Hör et al., 2020*), we also conducted northern analysis using RNA extracted from WT or Δ*hfq* cells across growth in LB (*Figure 5B*). Similar to other base-pairing

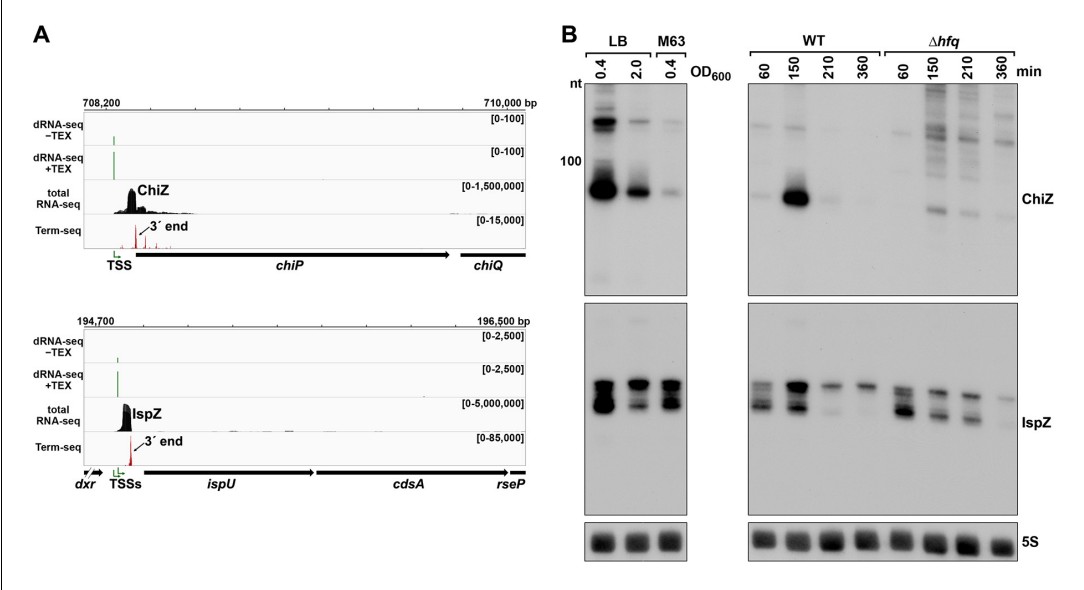

**Figure 5.** 5′ UTR-derived sRNAs ChiZ and IspZ. (**A**) RNA-seq screenshot of the ChiZ and IspZ loci displaying sequencing reads from the LB 0.4 growth condition from dRNA-seq (*Thomason et al., 2015*, HS2 samples), total RNA-seq and Term-seq. Total and Term-seq tracks represent an overlay of two biological replicates. Read count ranges are shown in the top of each frame. The chromosome nucleotide coordinates, relative orientation of the ORF (wide black arrow), dominant 3′ end from *Supplementary file 3* (small black arrow) and TSS (green bent arrow) as determined by the ratio of reads between ±TEX tracks, are indicated. (**B**) Northern analysis of ChiZ and IspZ. Left: the same WT (GSO988) RNA samples used for total RNA-seq and Term-seq in (**A**). Right: RNA was extracted from WT (GSO982) and Δ*hfq* (GSO954) cells at specific times after dilution of the overnight culture, (60, 150, 210, and 360 min) corresponding to early, middle, and late exponential and stationary phase. The Δ*hfq* strain reaches a lower final $OD_{600}$, yet exhibits a similar pattern of growth (*Melamed et al., 2020*). Total RNA was separated on an acrylamide gel, transferred to a membrane and probed for the indicated RNAs (RNAs were probed sequentially on the same membrane). The position of the 100 nt size marker is indicated for ChiZ (the region of the northern below 100 nt is shown for IspZ).

The online version of this article includes the following figure supplement(s) for figure 5:

**Figure supplement 1.** Predicted secondary structures and levels of 5′ derived ChiZ and IspZ sRNAs expressed from plasmids.

sRNAs, and consistent with Hfq binding, ChiZ abundance was low in the Δ*hfq* background. IspZ levels were only slightly affected by the absence of Hfq, even though this RNA is bound by Hfq (*Melamed et al., 2020*). Given that the binding of the sRNA ChiX to the mRNA *chiP* (containing ChiZ) increases Rho-mediated regulation of *chiP* (*Bossi et al., 2012*), we tested the role of Rho on ChiZ levels (*Figure 6A*). The effects of the *rhoR66S* mutant were dependent on growth phase, with a decrease in ChiZ for cells grown in LB to $OD_{600}$ ~0.4, when ChiZ levels are highest. In contrast, IspZ levels were not affected by the *rhoR66S* mutant (*Figure 2F*), and IspZ is likely subject to intrinsic termination as stated previously.

Given their association with Hfq, we tested the independent functions of ChiZ and IspZ as base-pairing sRNAs. Since the region of *chiP* encoding ChiZ has been documented to be a target for base pairing with the sRNA ChiX through compensatory mutations (*Figueroa-Bossi et al., 2009*; *Overgaard et al., 2009*), we postulated that ChiZ reciprocally regulates ChiX, sponging its base-pairing activity. To test this possibility, we assayed the effects of ChiZ overexpression. As for chromosomally-encoded ChiZ (*Figure 5B*), longer transcripts were observed for plasmid-encoded ChiZ, likely due to readthrough, but only the levels of the 90 nt ChiZ band were strongly reduced in the Δ*hfq* mutant (*Figure 5—figure supplement 1C*). Upon ChiZ overexpression in the WT background, we observed increased levels of the *chiP* mRNA, with a reciprocal change in ChiX levels (*Figure 6B*). The levels of the *chiP* mRNA overall were higher in the Δ*hfq* mutant background, but we no longer observed an increase upon ChiZ overexpression, likely due to the instabilities of ChiX and ChiZ. We also observed upregulation of a $P_{BAD}$-*chiP-lacZ* chromosomal translational fusion (*Schu et al., 2015*) upon ChiZ overexpression in a WT but not a Δ*chiX* background (*Figure 6C*). These observations

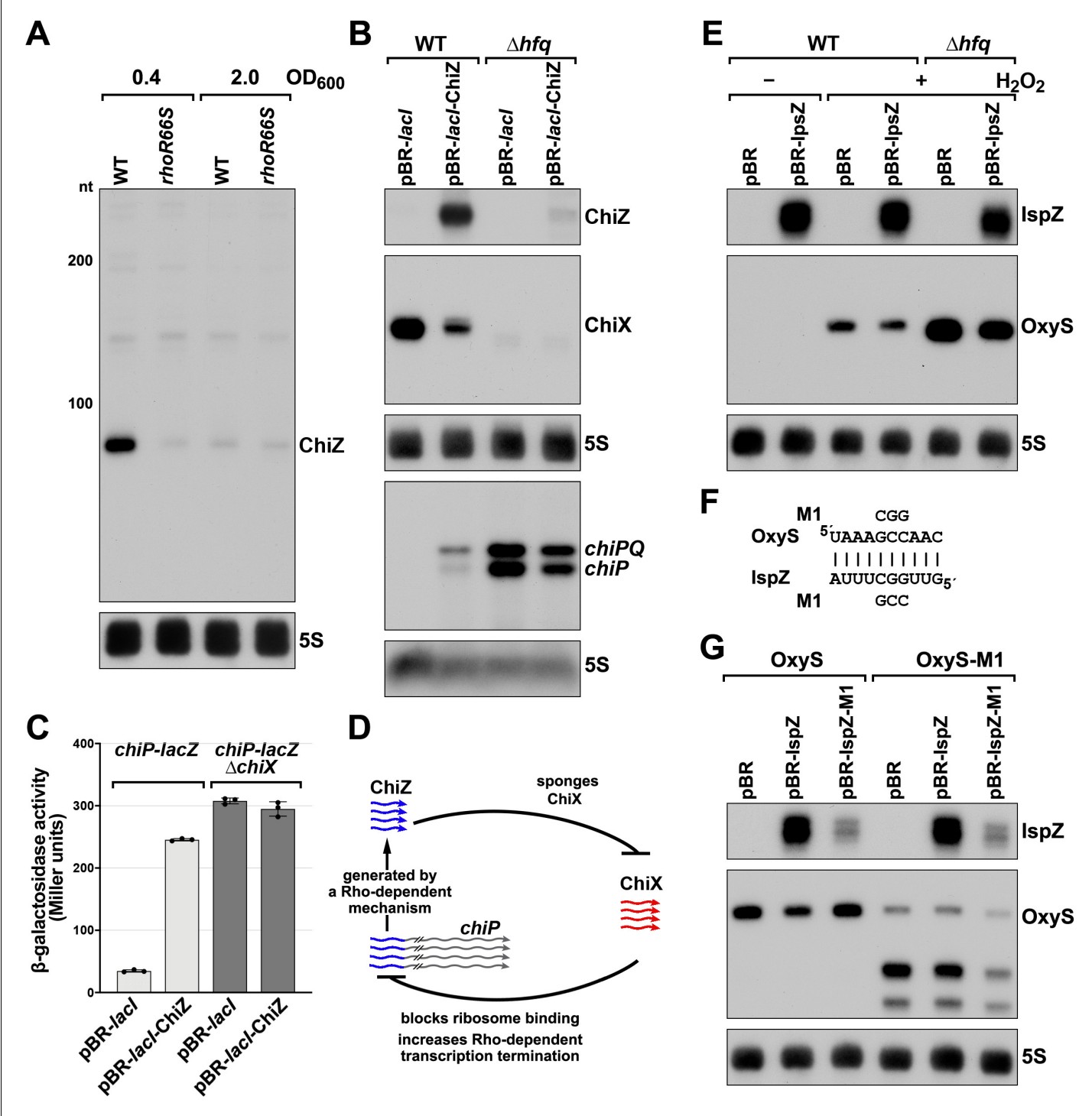

**Figure 6.** 5′ UTR-derived sRNAs ChiZ and IspZ act as sRNA sponges. (**A**) Northern analysis for ChiZ in WT (GSO989) and *rhoR66S* mutant (GSO990) cells. Cells were grown to $OD_{600}$ ~0.4 or 2.0 after a dilution of the overnight culture. Total RNA was extracted, separated on an acrylamide gel, transferred to a membrane and probed for ChiZ and 5S. This is the same blot depicted in *Figure 2F* and *Figure 2—figure supplement 1C*. (**B**) Northern analysis of ChiZ effect on *chiP* mRNA. RNA was extracted from WT (GSO982) and Δ*hfq* (GSO955) cells at 150 min after dilution of the overnight culture. Total RNA was separated on an acrylamide or agarose gel, transferred to a membrane and probed for the indicated RNAs (RNAs were probed sequentially on the same membrane). (**C**) β-galactosidase activity for *chiP* translational fusions to *lacZ* in WT (DJS2979) and Δ*chiX* (DJS2991) strains. Cells were grown and assayed 150 min after dilution of the overnight culture (see Materials and methods for details). Values represent the mean of three independent replicates (indicated by black dots). Error bars represent one standard deviation from the mean. (**D**) Model of ChiZ effects on ChiX, with indirect effects on the *chiP* mRNA. ChiZ (derived from the 5′ end of *chiP*) is blue and ChiX is red. (**E**) Northern analysis of IspZ

*Figure 6 continued on next page*

*Figure 6 continued*

effect on OxyS upon oxidative stress. WT (GSO982) and Δ*hfq* (GSO955) cells were grown for 150 min after dilution of the overnight culture, and WT (-H$_2$O$_2$) samples were collected. To induce OxyS, 0.2 mM H$_2$O$_2$ was spiked into the cultures for 20 min, and WT and Δ*hfq* samples were collected. Total RNA was extracted and separated on an acrylamide gel, transferred to a membrane and probed for the indicated RNAs (RNAs were probed sequentially on the same membrane). (F) Predicted base pairing between IspZ and OxyS with mutations assayed. (G) Test of direct interaction between IspZ and OxyS. RNA was extracted from WT (GSO982) and *oxyS*-M1 (GSO996) cells transformed with the pBR plasmids at 150 min after dilution of the overnight culture and 20 min incubation with 0.2 mM H$_2$O$_2$. Northern analysis was performed on total RNA as in (E). A smaller processed product was detected for the chromosomal OxyS-M1mutant, possibly due to secondary structural changes brought about by the M1 mutation.

support a novel sRNA regulatory network in which an mRNA (*chiP*) that is the target of an sRNA (ChiX) produces an RNA fragment (ChiZ) that reciprocally sponges the sRNA (ChiX) (*Figure 6D*).

We expected IspZ also might function as a base-pairing sRNA, and thus searched for potential targets identified by RIL-seq (*Melamed et al., 2020*; *Melamed et al., 2016*), an approach where RNAs in proximity on an RNA-binding protein are identified by co-immunoprecipitation, ligation, and sequencing of the chimeras. The predominant target for IspZ in these datasets is the oxidative stress-induced sRNA OxyS (*Altuvia et al., 1997*). This observation led us to test whether IspZ might act as a sponge of OxyS. As for chromosomally-encoded IspZ (*Figure 5*), and in contrast to ChiZ, little readthrough and no effect of Δ*hfq* was observed for plasmid-expressed IspZ (*Figure 5—figure supplement 1D*). We examined the effect of IspZ overexpression on OxyS levels in cells treated with 0.2 mM hydrogen peroxide, a condition known to induce OxyS expression (*Altuvia et al., 1997*). High levels of IspZ were associated with slightly lower OxyS levels, in line with sponging activity (*Figure 6E*). To obtain evidence for direct base pairing between IspZ and OxyS, IspZ was mutated on the overexpression plasmid (IspZ-M1), and compensatory mutations were introduced into the chromosomal copy of OxyS (OxyS-M1). Consistent with the predicted pairing (*Figure 6F*), IspZ-mediated down-regulation was eliminated with IspZ-M1 or OxyS-M1 alone but was restored when both mutations were present (*Figure 6G*).

## Putative ORF-internal sRNAs

We also noted examples of abundant 3′ ends internal to ORFs (*Supplementary file 3*), downstream of nearby 5′ ends previously identified by dRNA-seq (*Thomason et al., 2015*), and associated with a strong signal in total RNA-seq (*Figure 7*). A previous study inferred from total RNA-seq data that some sRNAs might be derived from sequences internal to ORFs (*Dar and Sorek, 2018a*). To test whether we could detect defined transcripts for these internal (int) signals, we selected candidate RNAs derived from the *ftsI* (renamed FtsO), *aceK*, *rlmD*, *mglC*, and *ampG* ORFs for further investigation (*Figure 7A*). Analysis of dRNA-seq data (*Thomason et al., 2015*) suggested that the FtsO, *aceK* int and *ampG* int 5′ ends likely are generated by RNase processing of the overlapping mRNA, whereas *rlmD* int and *mglC* int likely are transcribed from promoters internal to the overlapping ORFs. In nearly all cases, the RNA 3′ ends are not predicted to be due to Rho-dependent transcription termination events (*Supplementary file 3*), strongly suggesting they are generated by RNase processing or, for *aceK* int, intrinsic termination (intrinsic terminator score = 5.9, *Supplementary file 3*).

Northern analysis was performed for these RNAs using the same RNA analyzed in the Term-seq experiment (*Figure 7B*); distinct bands were detected for all the RNAs tested. While FtsO and *ampG* int were relatively abundant under all growth conditions tested, *mglC* int and *rlmD* int were only expressed in cells grown in LB, and *aceK* int was most abundant in LB at OD$_{600}$ ~ 2.0. We also conducted northern analysis using RNA extracted from WT or Δ*hfq* cells across growth (*Figure 7B*). The *aceK* int transcript was strongly dependent upon *hfq*, whereas the other RNAs were unaffected by *hfq* deletion, though all five transcripts have been reported to co-immunoprecipitate with Hfq (*Melamed et al., 2020*; *Melamed et al., 2016*). Additionally, the ORF-internal sRNAs are found in chimeras with other putative mRNA and sRNA targets in the Hfq RIL-seq datasets (*Melamed et al., 2020*; *Melamed et al., 2016*): FtsO and RybB, *aceK* int and *ompF*, *rlmD* int and MicA, *mglC* int and ArcZ, *ampG* int and CyaR. Significant chimeras also were detected between *aceK* int and *gatY* in the RIL-seq data set for the ProQ RNA chaperone (*Melamed et al., 2020*). Collectively, these data suggest these ORF-internal transcripts have independent regulatory functions.

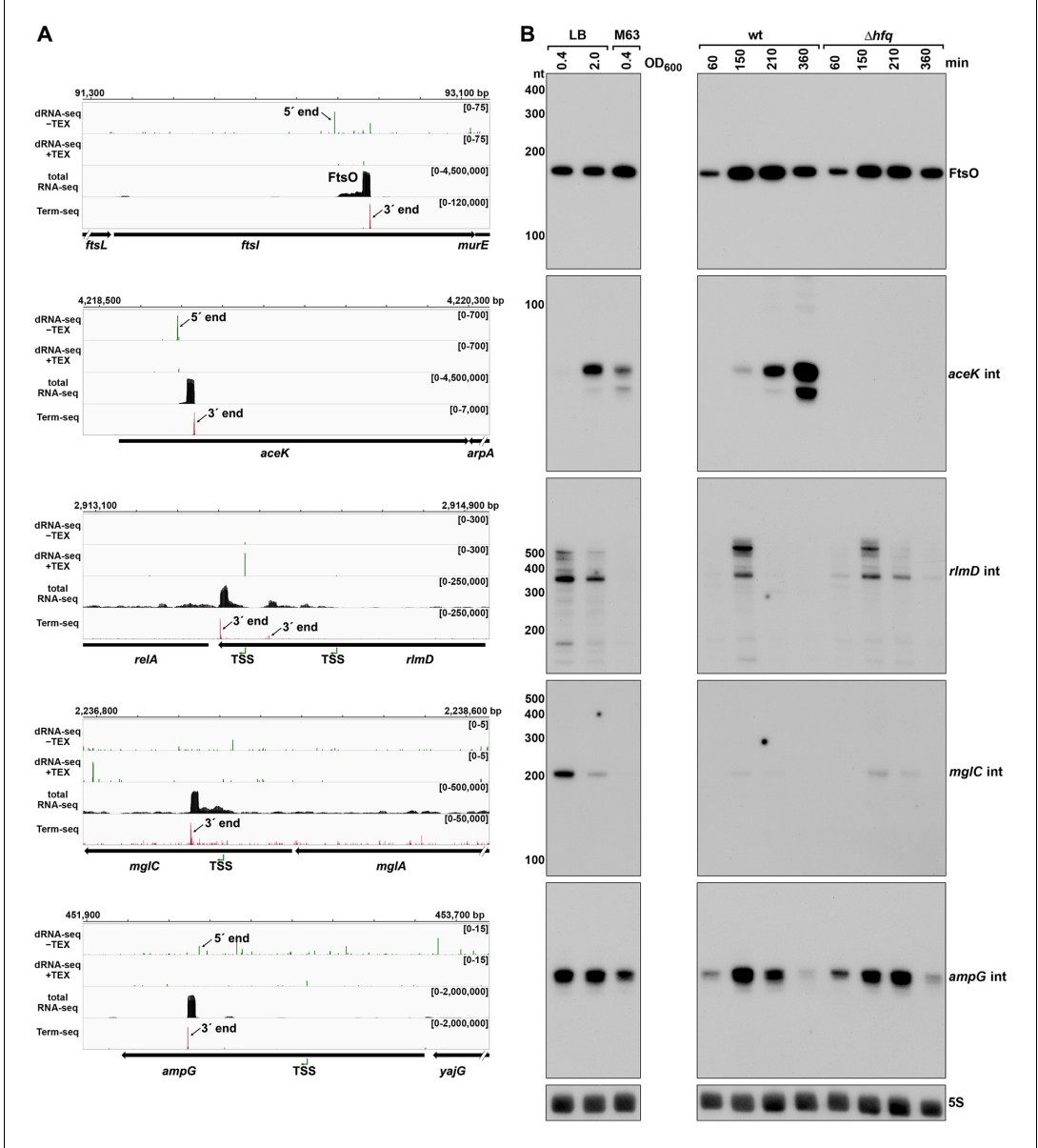

**Figure 7.** Detection of ORF-internal sRNAs. (**A**) RNA-seq screenshots of the *ftsI*, *aceK*, *rlmD*, *mglC*, and *ampG* mRNAs containing putative internal (int) sRNAs. Sequencing reads from the LB 0.4 dRNA-seq (***Thomason et al., 2015***, HS2 samples), total RNA-seq and Term-seq are displayed. Total RNA-seq and Term-seq tracks represent an overlay of two biological replicates. Read count ranges are shown in the top of each frame. The chromosome nucleotide coordinates, relative orientation of ORFs (wide black arrows), dominant 3′ end from ***Supplementary file 3*** (small black arrows labeled 3′ ends), and TSS (green bent arrows) or 5′ processed end (small black arrow labeled 5′ ends) as determined by the ratio of reads between ±TEX tracks, are indicated. (**B**) Northern analysis of ORF-internal sRNAs. Left: the same WT (GSO988) RNA samples used for total RNA-seq and Term-seq in (**A**); this is the same blot as depicted in ***Figure 5B***. Right: the same WT (GSO982) and Δ*hfq* (GSO954) RNA samples collected from cells at specific times after dilution of the overnight culture, (60, 150, 210, and 360 min) corresponding to early, middle, and late exponential and stationary phase as in ***Figure 5B*** (same blot with RNAs probed sequentially on the same membrane). Size markers are indicated for all RNAs (the region of the northern below 100 nt is shown for *ampG* int).

## ORF-internal FtsO is an sRNA sponge

To test for the suggested regulatory function, we focused on FtsO, which is encoded internal to the coding sequence of the essential cell division protein FtsI, and exhibited high levels across growth (***Figure 7B***). The predominant target for FtsO in the RIL-seq datasets was the sRNA RybB (***Figure 8A***) followed by the sRNA CpxQ (***Melamed et al., 2020***). The Hfq-mediated FtsO-RybB

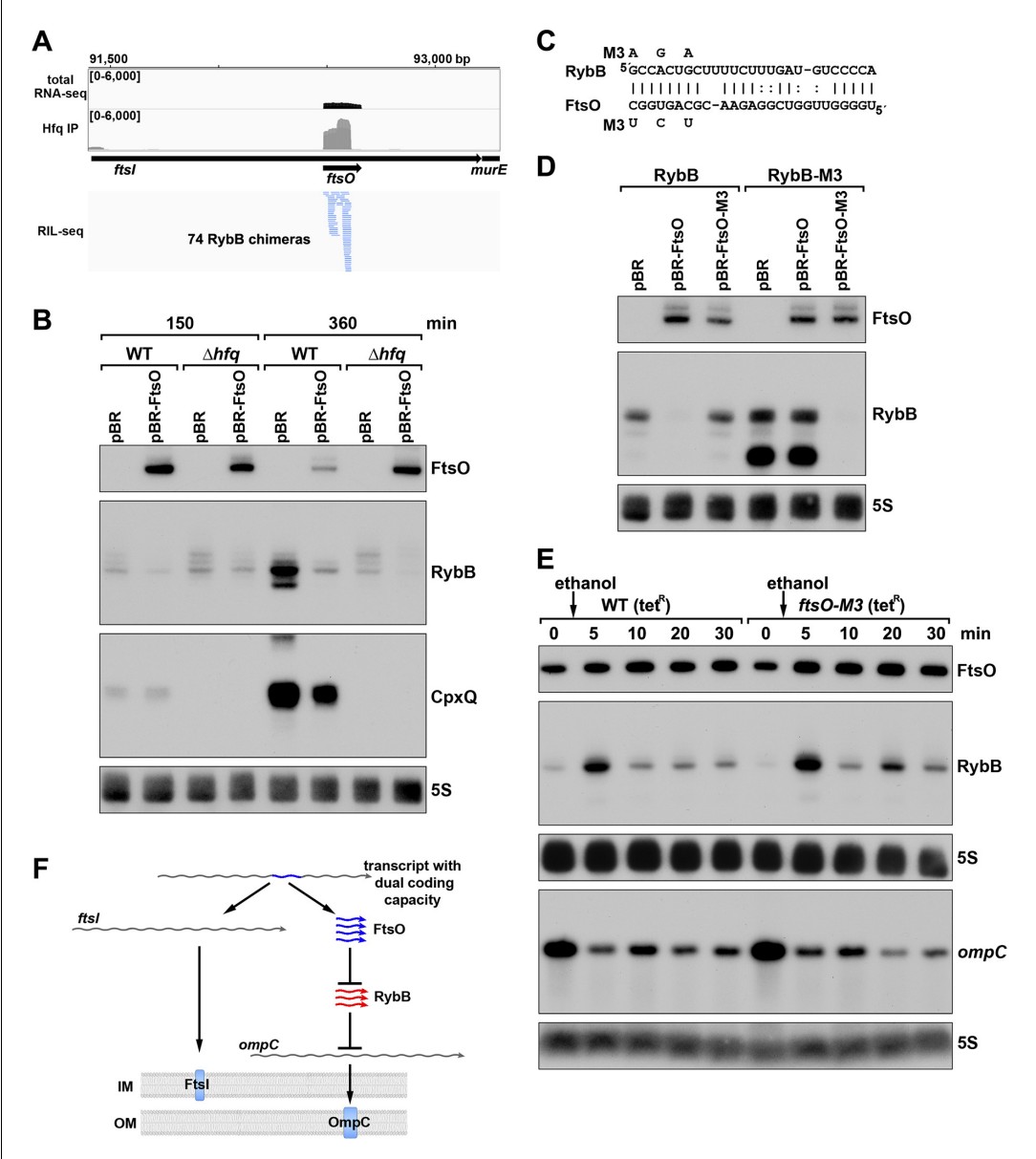

**Figure 8.** ORF-internal sRNA FtsO acts as a sponge of the RybB sRNA. (A) RIL-seq screenshot showing RybB chimeras at the *ftsO* locus. Data are from Hfq-FLAG LB RIL-seq performed 150 min after a dilution of the overnight culture, (***Melamed et al., 2020***, RIL-seq experiment 2). Top: signals for total RNA (dark gray) and Hfq RIL-seq single fragments with two biological repeats overlaid (light gray). Read count ranges are shown in the upper left of each frame. Bottom: chimeras with FtsO, blue lines indicate FtsO is the second RNA in the chimera. (B) RybB and CpxQ levels decrease in the presence of FtsO. RNA was extracted from WT (GSO982) and Δ*hfq* (GSO954) cells transformed with the indicated pBR plasmids at 150 and 360 min after dilution of the overnight culture. Total RNA was separated on an acrylamide gel, transferred to a membrane and probed for the indicated RNAs (RNAs were probed sequentially on the same membrane). (C) Predicted base pairing between FtsO and RybB with mutations assayed. (D) Test of direct interaction between FtsO and RybB. RNA was extracted from WT (GSO982) and *rybB*-M3 (GSO997) cells transformed with the indicated pBR plasmids at 360 min after dilution of the overnight culture. Northern analysis was performed on total RNA as in (B). (E) Chromosomally-encoded FtsO mutant dysregulates RybB levels under membrane stress. WT tet[R] (GSO998) and *ftsO*-M3 tet[R] (GSO999) cells were grown for 120 min after dilution of the overnight culture prior to the addition of EtOH to a final concentration of 5%. Cells were collected and extracted for RNA at the indicated time points after addition of EtOH. Northern analysis was performed on total RNA separated on either acrylamide or agarose gels as in ***Figure 4B and C***. (F) Model showing how same DNA sequence can encode two different gene products. The *ftsI* mRNA encodes the essential FtsI protein, found in the inner membrane (IM). This transcript also encodes the FtsO sRNA (blue), which blocks the activity of the RybB sRNA (red), induced by cell envelope stress, to down-regulate the synthesis of outer membrane (OM) porins such as OmpC.

The online version of this article includes the following figure supplement(s) for figure 8:

**Figure supplement 1.** FtsO and RbsZ sponging induces *ompC* levels and RybB reciprocally affects FtsO.

*Figure 8 continued on next page*

*Figure 8 continued*

**Figure supplement 2.** Co-conservation of *ftsO* and RybB.

interaction was also detected in an independent CLASH dataset (*Iosub et al., 2020*). We hypothesized that FtsO functions as a sponge for RybB and CpxQ, which are induced by misfolded outer membrane proteins and inner membrane proteins, respectively, and down-regulate the corresponding classes of proteins (reviewed in *Hör et al., 2020*). This model was tested by overexpressing FtsO in WT or Δ*hfq* cells grown to exponential and/or stationary phase (150 and 360 min after subculturing). For the 360 min time point when the levels of both RybB and CpxQ are highest, a reduction was observed for both sRNAs in cells overexpressing FtsO (*Figure 8B*). As reported previously, the levels of RybB and CpxQ are significantly lower in the Δ*hfq* strain, though some FtsO-dependent downregulation of RybB is still detected. Interestingly, we also observed that RybB overexpression had the reciprocal effect of decreasing FtsO levels, though only at the 360 min timepoint (*Figure 8— figure supplement 1A*).

To confirm the direct interaction between FtsO and RybB, we mutated three nucleotides in the site of predicted base pairing (*Figure 8C*) on the plasmid copy of *ftsO*. This mutation (FtsO-M3) abolished the effect of FtsO overexpression on RybB levels (*Figure 8D*). The repressive effect was similarly abolished by mutating the predicted site of base pairing in the chromosomal copy of *rybB* but was restored by combining the complementary mutations in *ftsO* and *rybB* (*Figure 8D*).

To test the downstream effect of sponging RybB, we examined the levels of the known RybB mRNA-target *ompC* (*Johansen et al., 2006*) compared the effects of RybB to the effects of the previously characterized RybB sRNA sponge, RbsZ (*Melamed et al., 2020*; *Figure 8—figure supplement 1B*). Both FtsO and RbsZ overexpression led to a decrease in RybB levels and a concomitant increase in *ompC* mRNA levels, though the effect of RbsZ was stronger. With both sRNA sponges, the level of OmpC protein was not obviously changed, likely because of the high levels of this protein.

Finally, we mutated the chromosomal copy of *ftsO* to introduce the same nucleotide substitutions (*Figure 8C*) that are silent with respect to the FtsI amino acid sequence. WT cells and cells carrying these mutations were treated with ethanol, causing outer membrane stress, which is known to induce RybB (*Peschek et al., 2019*). In both strains, a transient increase in RybB levels was observed 5 min after ethanol addition (*Figure 8E*). In WT cells, RybB levels then were decreased for 30 min after ethanol treatment. By contrast, in cells with mutant *ftsO*, RybB levels again increased at 20 min following ethanol treatment. This effect was also observed in a second experiment (*Figure 8—figure supplement 1C*) that documented higher RybB levels up to 60 min following ethanol treatment in *ftsO-M3* cells. We also assessed the consequences of FtsO sponging RybB on the levels of *ompC* for the same RNA samples. In both experiments, the *ompC* mRNA levels decreased after cells were treated with ethanol but for the *ftsO-M3* mutant strain there was a further decrease at the 20 min time point (*Figure 8E*). Along with RbsZ and the 3′ETS$^{leuZ}$ tRNA fragment (*Lalaouna et al., 2015*), FtsO is the third documented sponge of RybB. It is possible the other sponges mask some of the effects of FtsO.

The conservation of *ftsO* was examined by aligning *ftsI* orthologs across 18 species of gammaproteobacteria, revealing a striking degree of *ftsO* conservation (91% average identity) at *ftsI* wobble positions compared to the entire *ftsI* mRNA wobble positions (67% average identity) (*Figure 8—figure supplement 2A*). The region of base pairing parallels the conservation of the RybB seed sequence and is predicted to be within an unstructured region of FtsO (*Figure 8—figure supplement 2B and C*). Collectively, our data suggest that FtsO base pairing with RybB is conserved, and lowers RybB levels and activity following outer membrane stress (*Figure 8F*). The work demonstrates stable regulatory sRNAs can be derived from sequences internal to ORFs such that the same DNA sequence encodes two different functional molecules.

## Discussion

### Widespread premature transcription termination

Through our transcriptome-wide mapping of 3′ ends and Rho-dependent termination, we uncovered extensive RNA-mediated regulation and sRNA regulators encoded by 5′ UTRs and internal to ORFs. Other studies have previously identified RNA 3′ ends and regions of Rho-dependent termination in *E. coli* (*Dar and Sorek, 2018b*; *Ju et al., 2019*; *Peters et al., 2012*; *Yan et al., 2018*). While there was considerable overlap between our work and these prior studies, there were also substantial differences. For example, reporter gene fusion data and northern analysis supported Rho termination of *sugE*, *cfa*, *cyaA*, *speA*, and *mdtJ*, of which, only *sugE* was detected in the previous genome-wide surveys (*Dar and Sorek, 2018b*; *Ju et al., 2019*; *Yan et al., 2018*). Some of the differences between studies can be attributed to differences in the *E. coli* strains, growth conditions, and methods used. Indeed, we previously found that small methodological differences have a large impact on the identification of mapped TSSs (*Thomason et al., 2015*). Like any RNA-seq method, a few 3′ ends also could be due to RNA degradation during library preparation. Nevertheless, our follow-up experiments confirmed the biological relevance of several 3′ ends in 5′ UTRs and internal to ORFs. Given that strain and growth conditions used for our Term-seq and total RNA-seq match those of the previous dRNA-seq analysis (*Thomason et al., 2015*) in which we identified TSSs and 5′ processed ends, the combined sets represent a valuable resource for examining the *E. coli* transcriptome (see Materials and methods for links to interactive browsers).

The majority of the 3′ ends that we identified were classified as 'internal' or 'orphan', most of which map within 5′ UTRs or internal to ORFs, and a significant number of which are predicted to be generated by premature transcription termination. This notion of widespread premature transcription termination has been underappreciated in other studies that detected RNA 3′ ends and Rho-mediated termination. It is generally not possible to identify the exact position of Rho termination due to post-transcriptional RNA processing. Nevertheless, our reporter assays showed that in most cases tested, Rho termination could be localized to the 5′ UTR, suggesting that modulation of Rut accessibility in 5′ UTRs could be a common mechanism of regulation.

### Multiple levels of regulation at 5′ UTRs

Presuming that many premature termination events are regulatory, we documented and characterized examples of novel, diverse regulatory events for several of the 3′ ends. Undoubtedly, additional unique regulatory mechanisms exist for many of the other 3′ ends. We propose that the identification of 3′ ends in 5′ UTRs and ORFs is an effective approach to discover novel regulatory elements. Classically, these regulators, such as riboswitches and attenuators, have been identified by serendipity, studies of individual genes, or searches for conserved RNA structures (reviewed in *Breaker, 2018*), but these approaches may miss regulatory RNA elements if the function of the downstream gene is unknown or the region is not broadly conserved. Given that Term-seq is a sensitive, relatively unbiased, and genome-wide approach, it is another means of obtaining evidence for regulation in 5′ UTRs. As an example, the Term-seq data showed that transcripts from the 5′ UTR of the *E. coli* glycerol facilitator *glpF* have different 3′ ends under LB and M63 growth conditions, which could be due to uncharacterized regulation. 3′ end-mapping applied to *E. coli* grown under other conditions or other bacterial species should lead to the characterization of many more regulators, particularly in organisms such as the Lyme disease pathogen *Borrelia burgdorferi* (*Adams et al., 2017*) that lack any known *cis*-acting RNA elements. Consistent with broad applicability, Term-seq previously led to the identification of known riboregulators in *Bacillus subtilis* and *Enterococcus faecalis*, and a novel attenuator in *Listeria monocytogenes* (*Dar et al., 2016*).

A number of 3′ ends in 5′ UTRs and ORFs were found to be associated with uORFs. Our characterization of the *mdtU* uORF suggests that regulation of premature *mdtJI* transcription termination occurs in response to ribosome stalling induced by polyamines. Two recent studies showed that the polyamine ornithine can stall ribosomes immediately upstream of the stop codon of the *speFL* uORF, affecting Rho binding and the secondary structure of the *speFL* mRNA (*Ben-Zvi et al., 2019*; *Herrero Del Valle et al., 2020*). Strikingly, conservation of MdtU is strongest at the C-terminus, and overlaps a region of the *mdtU* RNA that is predicted to base pair with the *mdtJI* ribosome binding site. Thus, a mechanism similar to the one found for *speFL* may regulate *mdtJI* induction by

spermidine, a hypothesis that deserves further study, together with other examples where 3′ ends are located downstream of uORFs.

We also documented three instances of 3′ ends that localized a short distance downstream of known *trans*-acting sRNA base-pairing sites. These 3′ ends could be generated by endonuclease processing as a result of sRNA base pairing, or could be due to protection against exonucleases as a result of sRNA pairing. An examination of sRNA base-pairing sites predicted by RIL-seq points toward other instances of this type of regulation. For example, Term-seq identified 3′ ends immediately downstream of the predicted sRNA base-pairing regions for the uncharacterized mRNA-sRNA interactions *rbsD*-ArcZ, *dctA*-MgrR, and *yebO*-CyaR detected by RIL-seq chimeras (*Melamed et al., 2020*). In all these instances, the 3′ ends could be a result of the sRNA regulatory effect, and in some cases, may result in the formation of a new sRNA, as we observed for ChiZ. In general, our data further illustrate the complex regulation that occurs once transcription has initiated.

## Generation of sRNAs from 5′ UTRs and ORF-internal sequences

Previous studies have shown that intergenic regions and mRNA 3′ UTRs are major sources of regulatory sRNAs, with a few characterized examples of sRNAs derived from 5′ UTRs, and no characterized ORF-internal sRNAs (reviewed in *Adams and Storz, 2020*). Our data document that 5′ UTRs and ORFs can indeed encode functional base-pairing sRNAs. However, our work also raises important questions, including the mechanisms by which 5′ UTR-derived and ORF-internal sRNAs are generated.

Given that sRNAs derived from 5′ UTRs only require the generation of a new 3′ RNA end (likely sharing a TSS with their cognate mRNA), and are not usually constrained by codon sequences, they could evolve rapidly. We documented the formation of several 5′ UTR fragments by *cis*-regulatory events. These by-products of regulation could obtain independent regulatory functions, as has been reported for a few riboswitches and attenuators (*DebRoy et al., 2014*; *Melior et al., 2019*; *Mellin et al., 2014*). The sRNA 3′ end can be formed by intrinsic termination or Rho-dependent termination and/or processing. Importantly, for the downstream mRNA to be expressed, there needs to be some transcriptional readthrough. RNA structure predictions strongly suggest the IspZ 3′ end is generated by intrinsic termination (*Supplementary file 3*) for which we observed very little readthrough. In contrast, the ChiZ 3′ end is generated by Rho-dependent termination with significant readthrough that would allow *chiP* expression. The mechanisms that drive formation of either 5′-derived sRNAs or the corresponding full-length mRNAs could be regulated and are an interesting topic for future work.

Less is known about both the 5′ and 3′ ends of the ORF-internal sRNAs. The ends might be generated by ORF-internal promoters, termination, or RNase processing. In cases where one or both sRNA ends are generated by processing, this is presumably coupled with down-regulation of the overlapping mRNA. Strikingly, the number of sequencing reads for FtsO is orders of magnitude higher than that for the *ftsI* mRNA. How and when FtsO is produced are interesting questions for future studies. It is possible that the *ftsI* mRNA is protected from cleavage by ribosomes during cell division such that FtsO is only generated in the absence of mRNA translation. Other coding sequence-derived sRNAs, such as the putative regulator internal to *mglC*, likely have their own TSS. In some cases, such as *rlmD* int and *ampG* int, a transcript originating from a TSS internal to the cognate mRNA could be processed to form the 5′ end of the sRNA. These observations underscore how the interplay of transcription initiation, transcription termination, and RNase processing leads to many short transcripts that have the potential to evolve independent regulatory functions.

## Roles of 5′ UTR-derived and ORF-internal sRNAs

We identified and characterized three sRNA sponges that have 3′ ends either in 5′ UTRs or internal to the coding sequence. The first example, ChiZ-ChiX-*chiP*, represents a novel reciprocal autoregulatory loop. ChiZ is generated from the *chiP* 5′ UTR encompassing the site of pairing with the ChiX sRNA. We found ChiZ is formed by Rho-dependent termination, and in the absence of ChiX base pairing with *chiP*. When cells utilize chitobiose as a carbon source and ChiX levels are naturally low, there are higher levels of *chiP* (*Overgaard et al., 2009*) and likely also higher levels of ChiZ. In this model, when chitooligosaccharides need to be imported, ChiZ prevents ChiX from base pairing, and promotes degradation of ChiX. When metabolic needs shift, the levels of ChiZ could decrease,

allowing ChiX to regulate *chiP* and other targets. This may work in competition, conjunction, or at separate times from the *chbBC* intergenic mRNA sequence, which also sponges and promotes decay of ChiX (*Overgaard et al., 2009*). It will be interesting to see if other 5´ UTRs and sRNAs form similar autoregulatory loops, since 5´ UTRs are enriched for sRNA pairing sites, and we have shown that sRNA pairing is associated with distinct small transcripts from 5´ UTRs.

In a second example, we characterized IspZ, which is generated from the 5´ UTR of *ispU* (*uppS*), encoding the synthase for undecaprenyl pyrophosphate (UPP), a lipid carrier for bacterial cell wall carbohydrates (*Apfel et al., 1999*). We suggest IspZ may connect cell wall remodeling with the oxidative stress response. Cellular levels of toxic reactive oxygen species are increased when cell wall synthesis is blocked, and oxidative damage impedes *ispU*-related cell wall growth (*Kawai et al., 2015*). Thus, IspZ downregulation of the hydrogen peroxide-induced sRNA OxyS may dampen the oxidative response at a time when the response might be detrimental.

While small transcripts from within coding sequences have been noted previously (*Dar and Sorek, 2018a*; *Reppas et al., 2006*), and a homolog of FtsO (STnc475) has been detected for *Salmonella enterica* (*Smirnov et al., 2016*), we are the first to document a regulatory role for a bacterial ORF-internal sRNA. FtsO was found to base pair with, and negatively regulate the membrane stress response sRNA, RybB. The *ftsI* mRNA encodes a low-abundance but essential penicillin-binding protein that is localized to the inner membrane at the division site and cell pole (*Weiss et al., 1997*). The cell may need to alter its response to membrane stress during the division cycle when many membrane components are needed, and we suggest FtsO could facilitate crosstalk between cell division and membrane stress by regulating RybB activity. Intriguingly, we observed the greatest effect of ethanol addition on RybB levels in the Δ*ftsO* background at 20 min, which is also the doubling time for *E. coli* MG1655. While it is reasonable to assume that regulatory sRNAs encoded by intragenic sequences are rare, due to the challenge of encoding two functions in one region of DNA, we think it is likely that other ORF-internal sRNAs have function.

Our work has significantly increased the number of sRNAs documented to modulate the activities of other sRNAs by sponging their activities, as found for ChiZ, or affecting their levels, as shown for IspZ and FtsO. The findings raise other questions including how many more short transcripts generated by termination or processing have regulatory functions. It also is intriguing that some abundant sRNAs are subject to regulation by multiple sponges, including ChiX, which is regulated by ChiZ and the *chbBC* intergenic region (*Overgaard et al., 2009*), and RybB, which is regulated by FtsO, RbsZ (*Melamed et al., 2020*) and the 3´ETS$^{leuZ}$ tRNA fragment (*Lalaouna et al., 2015*). Finally, little is known about how the activities and levels of the sponges themselves are regulated. The levels of some, but not others, are influenced by Hfq binding. A number appear to be constitutively expressed, such that target sRNA levels must increase to overcome the effects of the sponges.

Further identification and characterization of RNA fragments generated by premature termination or processing, detected by mapping the 5´ and 3´ ends of bacterial transcriptomes, will help elucidate the effects of regulatory RNAs. Our datasets point to a plethora of potential *cis*- and *trans*-acting regulatory elements in 5´ UTRs and ORF-internal regions, providing a valuable resource for further studies of gene regulation.

# Materials and methods

## Key resources table

| Reagent type (species) or resource | Designation | Source or reference | Identifiers | Additional information |
|---|---|---|---|---|
| Strain, strain background (*Escherichia coli*) | MG1655 (WT) | this study | N/A | see *Supplementary file 4* for derivatives |
| Antibody | mouse monoclonal anti-FLAG-M2-HRP | Sigma-Aldrich | Cat#A8592 | WB (1:2000) |
| Antibody | rabbit polyclonal anti-OmpC | Biorbyt | Cat#orb6940 | WB (1:500) |

*Continued on next page*

*Continued*

| Reagent type (species) or resource | Designation | Source or reference | Identifiers | Additional information |
|---|---|---|---|---|
| Antibody | donkey polyclonal peroxidase labeled anti-rabbit | GE Healthcare | Cat#NIF824 | WB (1:1000) |
| Recombinant DNA reagent | pBRplac | *Guillier and Gottesman, 2006* | N/A | see *Supplementary file 4* for derivatives |
| Recombinant DNA reagent | pNM46 (pBRplac-*lacI*) | this study | N/A | see *Supplementary file 4* for derivatives |
| Recombinant DNA reagent | pKD46 | *Datsenko and Wanner, 2000* | N/A | see *Supplementary file 4* for derivatives |
| Recombinant DNA reagent | pMM1 | *Stringer et al., 2014* | N/A | see *Supplementary file 4* for derivatives |
| Sequence-based reagent | northern probes and primers | this study | N/A | see *Supplementary file 4* |
| Chemical compound, drug | spermidine | Sigma-Aldrich | Cat#S2626-1G | |
| Other | TRIzol reagent | Thermo Fisher Scientific | Cat#15596018 | RNA extractions |
| Other | RNA-sequencing reagents | *Melamed et al., 2018* | N/A | |
| Other | ureagel-8 | National Diagnostics | Cat#EC-838 | acrylamide northern solution |
| Other | ureagel complete | National Diagnostics | Cat#EC-841 | acrylamide northern solution |
| Other | NuSieve 3:1 agarose | Lonza | Cat#50090 | agarose for northern blotting |
| Other | 37% formaldehyde | Fisher Scientific | Cat#BP531-500 | |
| Other | RiboRuler high range RNA ladder | Thermo Fisher Scientific | Cat#SM1821 | |
| Other | RiboRuler low range RNA ladder | Thermo Fisher Scientific | Cat#SM1831 | |
| Other | Zeta-Probe blotting membrane | Bio-Rad | Cat#1620159 | northern membrane |
| Other | ULTRAhyb-oligo hybridization buffer | Thermo Fisher Scientific | Cat#AM8663 | |
| Other | $\gamma$-$^{32}$P ATP | PerkinElmer | Cat#NEG035C010MC | |
| Other | T4 polynucleotide kinase | New England Biolabs | Cat#M0201L | |
| Other | Illustra MicroSpin G-50 columns | GE Healthcare | Cat#27533001 | |
| Other | mini-PROTEAN TGX gels | Bio-Rad | Cat#456–1086 | |
| Other | EZ-RUN pre-stained Rec protein ladder | Fisher Scientific | Cat#BP3603-500 | |
| Other | nitrocellulose membrane | Thermo Fisher Scientific | Cat#LC2000 | western membrane |
| Other | SuperSignal West Pico PLUS chemiluminescent substrate | Thermo Fisher Scientific | Cat#34580 | |
| Software, algorithm | lcdb-wf | 'lcdb-wf'. Dale et al., GitHub Repository | v1.5.3 | https://github.com/lcdb/lcdb-wf |
| Software, algorithm | sra-tools | SRA Toolkit Development Team | v2.9.1_1 | http://ncbi.github.io/sra-tools/ |
| Software, algorithm | cutadapt | *Martin, 2011* | v2.3 | https://github.com/marcelm/cutadapt |
| Software, algorithm | fastqc | *Wingett and Andrews, 2018* | v0.11.8 | https://qubeshub.org/resources/fastqc |
| Software, algorithm | bwa | *Li and Durbin, 2010* | v0.7.17 | http://bio-bwa.sourceforge.net/ |

*Continued on next page*

*Continued*

| Reagent type (species) or resource | Designation | Source or reference | Identifiers | Additional information |
|---|---|---|---|---|
| Software, algorithm | bowtie2 | *Langmead and Salzberg, 2012* | v2.3.5 | http://bowtie-bio.sourceforge.net/bowtie2 |
| Software, algorithm | samtools | *Li et al., 2009* | v1.9 | http://www.htslib.org/ |
| Software, algorithm | subread | *Liao et al., 2013* | v1.6.4 | http://subread.sourceforge.net/ |
| Software, algorithm | multiqc | *Ewels et al., 2016* | v1.7 | https://github.com/ewels/MultiQC |
| Software, algorithm | picard | 'Picard Toolkit.' 2019. Broad Institute, GitHub Repository | v2.20.0 | https://github.com/broadinstitute/picard |
| Software, algorithm | deeptools | *Ramírez et al., 2016* | v3.2.1 | https://deeptools.readthedocs.io/ |
| Software, algorithm | termseq-peaks | 'termseq-peaks.' 2020. NICHD-BSPC, GitHub Repository | N/A | https://github.com/NICHD-BSPC/termseq-peaks |
| Software, algorithm | bedtools | *Quinlan and Hall, 2010* | v2.27.1 | https://bedtools.readthedocs.io/ |
| Software, algorithm | pybedtools | *Dale et al., 2011* | v0.8.0 | https://github.com/daler/pybedtools |
| Software, algorithm | ucsc-toolkit | *Kent et al., 2010* | v357 | https://doi.org/10.1093/bioinformatics/btq351 |
| Software, algorithm | biopython SeqIO | *Cock et al., 2009* | v1.73 | biopython.org |
| Software, algorithm | rhoterm-peaks | 'rhoterm-peaks' 2020. GitHub repository | N/A | https://github.com/gbaniulyte/rhoterm-peaks |
| Software, algorithm | CLG Genomics Workbench | Qiagen | v8.5.1 | alignment of Direct RNA-seq reads |
| Software, algorithm | CLUSTAL | *Madeira et al., 2019* | v2.1 | https://www.ebi.ac.uk/Tools/msa/clustalo/ |
| Software, algorithm | BOXSHADE | | v3.21 | https://embnet.vital-it.ch/software/BOX_form.html |
| Software, algorithm | Sfold | *Ding et al., 2004* | v2.2 | http://sfold.wadsworth.org/cgi-bin/srna.pl |

## Bacterial strains and plasmids

Derivatives of *E. coli* K12 MG1655 (WT) were used for all experimental studies. All strains, plasmids and oligonucleotides used are listed in *Supplementary file 4*. Engineered mutations and plasmid inserts were verified by sequencing.

The *E. coli* strains carrying *mdtJ*-3XFLAG were engineered using the FRUIT method (*Stringer et al., 2012*). Briefly, a 3XFLAG-*thyA*-3XFLAG tag/selection marker was PCR-amplified with primers JW9000 + JW9001 and recombineered into strain AMD061 and then counter-selected for *thyA* loss due to recombination of the FLAG tags, resulting in the intermediate strain YY18. This intermediate was used to create the *mdtU* start codon mutant (ATG→ACG) by recombineering the PCR-amplified *thyA* marker (using primers JW10309 + JW10310) into the *mdtU* gene and then replacing the *thyA* marker with the *mdtU* mutation (recombineering a PCR-amplified product made using primers JW10311 - JW10314). The native *thyA* locus was restored as described previously (*Stringer et al., 2012*). This resulted in the wild-type *mdtU mdtJ*-3XFLAG (YY20) or the *mdtU* start codon (ATG→ACG) mutant *mdtJ*-3XFLAG (AMD742) strain. The Δ*mgrR*::kan (GSO769), Δ*micA*::kan (GSO157), and Δ*chiX*::kan (GSO169) deletions (*Hobbs et al., 2010*) were transduced into MG1655 (GSO982) by P1 transduction, resulting in GSO993, GSO994 and GSO995 respectively. The *oxyS*-M1::kan (GSO996) and *rybB*-M3::kan (GSO997) strains were constructed by PCR-amplifying the kan$^R$ sequence in pKD4 (*Datsenko and Wanner, 2000*) using primers PA313 + PA314 (*oxyS*-M1) or

PA218 + PA219 (rybB-M3) and recombineering the product (*Datsenko and Wanner, 2000*; *Yu et al., 2000*) into the chromosome of *E. coli* NM400 (kind gift of Nadim Majdalani). The *ftsO*-M3::kan (GSO999) strain was constructed by first transforming a temperature sensitive *ftsI* Y380D mutant (PA215) (*Dai et al., 1993*), with pKD46 (*Datsenko and Wanner, 2000*). This strain was electroporated with an *ftsO*-M3 PCR product (amplified from the *ftsO*-M3 geneblock (*Supplementary file 4*) with primers PA216 + PA217) in the presence of 20 mM L-arabinose to induce the λ recombinase on pKD46. Colonies were selected by plating at 45 ˚C. Colony PCRs (using primers PA216 + PA217) and sequencing was performed to check for repair of the *ftsI* Y380D mutation and simultaneous *ftsO*-M3 incorporation (GSO999). A colony without the *ftsO*-M3 change was kept as a wild-type control (GSO998). All sRNA mutant alleles were subsequently transferred into *E. coli* MG1655 (GSO982) by P1 transduction.

The pMM1 β-galactosidase transcriptional reporter fusion plasmid was constructed by PCR-amplifying the high expression promoter KAB-TTTG (*Burr et al., 2000*) from pJTW064 (*Stringer et al., 2014*) using primers JW10252 + JW10253, and the DNA was ligated into pAMD-BA-lacZ plasmid (*Stringer et al., 2014*) digested with the *Nsi*I and *Hind*III restriction enzymes. Either the entire 5´ UTR and annotated ORF region, or 5´ UTR region alone for selected genes was PCR-amplified using the oligonucleotides listed in *Supplementary file 4*, and cloned into the pMM1 vector, cut with *Nsi*I and *Nhe*I restriction enzymes, using the NEBuilder HiFi kit (NEB). The *mdtU-lacZ* translational fusions were constructed by PCR-amplifying the *mdtU* gene from either a WT (*E. coli* MG1655) genomic template or *mdtU* start codon mutant (ATG→ACG) template (AMD742) using primers JW7269 + JW8934. These PCR products were subsequently ligated into the pAMD-BA-*lacZ* plasmid (*Stringer et al., 2014*), cut with *Sph*I and *Hind*III restriction enzymes, using the NEBuilder HiFi kit (NEB). sRNAs were over expressed using the pBRplac plasmid (*Guillier and Gottesman, 2006*). sRNA sequences were PCR-amplified using the oligonucleotides listed in *Supplementary file 4*, digested with *Aat*II and *Eco*RI, and cloned into pBRplac digested with the same restriction enzymes. The ChiZ overexpression construct was engineered using the NEBuilder kit (NEB), according to the manufacturer's instructions with primers PA311 + PA312 and LW043 + LW044 and the pBRplac-*lacI* derivative, pNM46 (kind gift of Nadim Majdalani).

## Growth conditions

Bacterial strains standardly were grown with shaking at 250 rpm at 37 ˚C in either LB rich medium or M63 minimal medium supplemented with 0.2% glucose and 0.001% vitamin B1. Ampicillin (100 μg/ml), kanamycin (30 μg/ml), chloramphenicol (30 μg/ml) and/or IPTG (1 mM) were added where appropriate. Unless indicated otherwise, overnight cultures were diluted to an $OD_{600}$ of 0.05 and grown to the indicated $OD_{600}$ or time point.

## RNA isolation

*E. coli* cells corresponding to the equivalent of 10 $OD_{600}$ were collected by centrifugation, washed once with 1X PBS (1.54 M NaCl, 10.6 mM $KH_2PO_4$, 56.0 mM $Na_2HPO_4$, pH 7.4) and pellets snap frozen in liquid $N_2$. RNA was isolated using TRIzol (Thermo Fisher Scientific) exactly as described previously (*Melamed et al., 2020*). RNA was resuspended in 20–50 μl DEPC $H_2O$ and quantified using a NanoDrop (Thermo Fisher Scientific).

## Term-seq

Two biological replicates of *E. coli* MG1655 (GSO988) were diluted 1:500 from an LB overnight culture in either LB or M63 glucose media. Cells were collected at an $OD_{600}$ ~0.4 and 2.0 for LB and an $OD_{600}$ ~0.4 for M63 grown cultures. RNA was extracted as described above and analyzed using an Agilent 4200 TapeStation System to check the quality. Any contaminating DNA in the samples was removed by treating 15 μg of RNA with 10 U of DNase I (Roche) for 15 min at 37 ˚C in the presence of 80 U of recombinant RNase inhibitor (Takara Bio). Next, RNA was purified by mixing the sample with an equal volume of phenol stabilized:chloroform:isoamyl alcohol (25:24:1) and centrifugation at maximum speed in Heavy Phase Lock Gel tubes (5 PRIME). A volume of chloroform, equal to the original sample volume, was added to the same Heavy Phase Lock Gel tubes and spun again. The aqueous layer was removed and ethanol precipitated in the presence of 15 μg GlycoBlue (Ambion). RNA pellets were reconstituted in 10 μl DEPC $H_2O$ and analyzed using an Agilent 4200 TapeStation

System to ensure DNase-treated RNA was at high quality. Term-seq libraries were prepared using a modified version of the RNAtag-seq methodology (*Shishkin et al., 2015*), based on the previously published Term-seq methodology (*Dar et al., 2016*). 1.5 µg of DNA-free RNA was first ligated at the 3´ end with 150 µM barcoded oligonucleotide adapters which were 5´phosphorylated and dideoxycytidine 3´ terminated (*Supplementary file 4*). RNA and 3´ adapters were incubated at 22 °C for 2.5 hr with 51 U of T4 RNA Ligase I (NEB) and 12 U of recombinant RNase inhibitor (Takara Bio) in 1X T4 RNA Ligase Buffer (NEB), 9% DMSO, 20% PEG 8000, and 1 mM ATP. 3´ ligated RNA was cleaned by incubating with 2.5X volume of RNAClean XP beads (Beckman Coulter) and 1.5X volume of isopropanol for 15 min, before separation on a magnetic rack. Bead-bound RNA was washed with 80% ethanol, air dried, and resuspended in DEPC $H_2O$. RNA-containing-supernatants were removed and the same RNAClean XP bead cleanup protocol was repeated, with a final DEPC $H_2O$ elution of 9.5 µl. RNA was fragmented by incubating 9 µl of cleaned-up RNA with 1X Fragmentation Reagent (Invitrogen) for 2 min at 72 °C, followed by an addition of 1X Stop Solution (Invitrogen). Samples were stored on ice following individual fragmentation of each sample. Fragmented-RNA was pooled together and cleaned using the RNA Clean and Concentrator-5 kit (Zymo) according to the manufacturer's instructions. Library construction continued following the bacterial-sRNA adapted, RNAtag-seq methodology starting at the rRNA removal step (*Melamed et al., 2018*). Term-seq RNA libraries were analyzed on a Qubit 3 Fluorometer (Thermo Fisher Scientific) and an Agilent 4200 TapeStation System prior to paired-end sequencing using the HiSeq 2500 system (Illumina).

## Identification of 3´ ends from Term-seq

Raw sequence reads were processed using lcdb-wf (lcdb.github.io/lcdb-wf/) according to the following steps. Raw sequence reads were trimmed with cutadapt 1.18 (*Martin, 2011*) to remove any adapters while performing light quality trimming with parameters '-a AGATCGGAAGAGC -q 20 –minimum-length = 25.' Sequencing library quality was assessed with fastqc v0.11.8 with default parameters. The presence of common sequencing contaminants was evaluated with fastq_screen v0.11.3 with parameters '–subset 100000 –aligner bowtie2.' Trimmed reads were mapped to the *E. coli* reference genome (MG1655 NC_000913.3) using BWA-MEM. Multimapping reads were filtered using samtools (*Li et al., 2009*). Uniquely aligned reads were then mapped to gene features using subread featureCounts v1.6.2 with default parameters. BedGraph files were generated using deepTools (*Ramírez et al., 2016*) on reads from each strand separately.

An initial set of termination peaks was called per sample on the bedGraph files from uniquely aligned reads using a novel signal processing approach combined with a statistically-informed method of combining multiple replicates. Briefly, the scipy.signal Python package was used to call peaks on each replicate in a manner which handled high, sharp peaks as found in Term-seq data, using the scipy.signal.find_peaks function with a width of (1, None), a prominence of (None, None), and a relative height of 0.75. Peaks for each replicate were then combined using the IDR framework (*Landt et al., 2012*) into a set of peaks that were reproducible (both strong and consistent) across replicates. The code for this can be found at https://github.com/NICHD-BSPC/termseq-peaks *Adams, 2020* (copy archived at swh:1:rev:8bb48d2a034b22a312a7e848e0f8694a284a8324) and can be used in general for Term-seq peak-calling in other bacteria. Termination peaks were subsequently curated according to the following criteria. The single-bp peak coordinate was set to the strongest signal nucleotide within the boundary of the initial broader peak using multiBigWigSummary from deepTools 3.1.3. The most downstream position, relative to the peak orientation, was chosen when several positions were equally strong. Scores from peaks within a distance of up to 100 bp were assessed to select the peak with the highest score among the cluster for further analysis. These curated peaks were used for all analysis herein (*Supplementary file 1*).

## Total RNA-seq

Total RNA-seq was performed using the same RNA that was used for the Term-seq library preparations. Total RNA-seq library construction was carried out based on the RNAtag-seq methodology (*Shishkin et al., 2015*), which was adapted to capture bacterial sRNAs (*Melamed et al., 2018*). Total RNA-seq RNA libraries were sequenced as for Term-seq. Total RNA-seq data processing followed the same procedures as Term-seq data analysis for QC, adaptor removal and sequencing read mapping.

## BCM treatment and DirectRNA-seq

One culture of *E. coli* MG1655 cells (GSO989) was grown in LB to an $OD_{600}$ ~0.5 and the culture was split and half was treated with 100 μg/ml of BCM (gift from Max Gottesman) for 15 min. Total RNA was isolated from 1.5 ml of untreated and BCM-treated cultures using the hot-phenol RNA extraction method followed by ethanol precipitation as described previously (*Stringer et al., 2014*). Genomic DNA was removed by treating 8 μg of total RNA with 4 U of Turbo DNase (Invitrogen) for 45 min at 37 °C. DNA-free RNA was purified using phenol:chloroform:isoamyl alcohol and ethanol precipitation as described previously (*Stringer et al., 2014*). rRNA was removed using a Ribo-Zero (Bacteria) kit (Epicenter) according to the manufacturer's instructions. The RNA libraries were prepared and processed at the Helicos BioSciences facility where poly-A tails and a 3′-dATP block were added to make the RNA suitable for direct sequencing on the HeliScope Single-Molecule Sequencer (*Ozsolak and Milos, 2011*).

## Identification of Rho-dependent 3′ ends

CLC Genomics Workbench (v8.5.1) was used to align DirectRNA-sequencing reads from untreated and BCM-treated samples to the MG1655 NC_000913.3 sense and reverse-complemented genome to properly identify the position of the first mapped 3′ end nucleotide. Mapping parameters were set to default, except for 'Length fraction' and 'Similarity fraction', which were set to 0.7 and 0.9, respectively. Quality scores were not generated by the HeliScope Sequencer; arbitrary quality scores were added to each read in fasta files to fit import requirements for CLC Genomics Workbench, but they were ignored when mapping. The read count and position of sequenced transcript 3′ ends were used for further analysis. The approximate Rho-dependent transcription termination sites were predicted by identifying the locations of transcriptional readthrough in the BCM-treated sample. The read counts in 800 nt regions upstream (BCM_us) and downstream (BCM_ds) of each position were compared to read counts from the same positions in the untreated sample (Untreated_us, Untreated_ds). We then used a Fisher's exact test to compare the downstream:upstream read count ratios for untreated and BCM-treated samples, and we calculated a ratio of ratios: ($R_{(BCM/untreated)} = \frac{BCM\_ds/BCM\_us}{Untreated\_ds/Untreated\_us}$). We refer to the *p*-value from the Fisher's exact test as the 'significance score', and we refer to the $R_{(BCM/Untreated)}$ ratio as the 'Rho score' (*Supplementary file 2*). Putative Rho termination regions were those genome coordinates with a positive Rho score and a significance score $<1e^{-4}$. Only the position with the highest Rho significance score within an 800 nt window upstream and downstream is reported in *Supplementary file 2*. Note that Rho scores and significance scores listed in *Supplementary file 3* were calculated for specific positions matching the dominant Term-seq 3′ ends; there may be nearby positions with a lower significance score and/or a higher Rho score.

## Classification of 3′ ends

The intersect function of Bedtools 2.28.0 (*Quinlan and Hall, 2010*), ran via pybedtools v0.8.0 (*Dale et al., 2011*), was used to assign each peak to one or more classes: Primary (3′ peaks located on the same strand either within 50 bp downstream of the 3′ end of an annotated mRNA ORF, tRNA, rRNA or sRNA with the highest score), Antisense (3′ peaks located on the opposite strand of an annotated mRNA ORF, tRNA, rRNA or sRNA within 50 bp of its start and end coordinates), Internal (3′ peaks located on the same strand within an annotated mRNA ORF, tRNA, rRNA or sRNA coordinates, excluding the 3′ end coordinate) and Orphan (3′ peak not falling in any of the previous classes).

3′ ends were also categorized according to their position relative to mRNA 5′ UTRs and internal mRNA regions (*Supplementary file 3*). Any 3′ end (*Supplementary file 1*) that was located within a region of 200 bp upstream of an annotated start codon to the stop codon were extracted and further analyzed. To remove any 3′ ends that likely belonged to an upstream gene in the same direction, TSS data (*Thomason et al., 2015*) obtained using the same growth conditions and *E. coli* strain as Term-seq was considered. All these 3′ ends were examined for the first upstream feature (either a TSS or an ORF stop codon). Any 3′ end where the first upstream feature was a stop codon was eliminated, unless there was also a TSS ≤200 bp upstream the 3′ end or that upstream feature was the stop codon of an annotated 'leader peptide' on the EcoCyc *E. coli* database (*mgtL, speFL, hisL, ivbL, ilvL, idlP, leuL, pheL, pheM, pyrL, rhoL, rseD, thrL, tnaC, trpL, uof*). Any 3′ end where a TSS was

only 20 bp or less upstream was also eliminated. This resulted in the 3′ end coordinates in *Supplementary file 3*. For the LB 0.4 condition, 3′ ends were given a Rho score from Direct-RNA-seq (as described above) and an intrinsic terminator score (with a custom script as defined in *Chen et al., 2013*). uORFs for which synthesis was detected by western analysis and/or translational reporter fusions (*Hemm et al., 2008*; *VanOrsdel et al., 2018*; *Weaver et al., 2019*), sRNAs for which synthesis was detected by northern analysis (this study) and other characterized RNA regulators were noted for the LB 0.4 condition.

## Comparison of Term-seq 3′ ends and Rho termination regions to equivalent genomic positions identified in other studies

As detailed in *Figure 1—figure supplement 2*, we compared the 3′ ends identified by Term-seq with 3′ ends identified in other studies, and we compared Rho termination regions with putative sites of Rho termination identified in other studies. Specifically, we determined how many of the positions in our list of genome coordinates are within a given distance threshold of a genome coordinate from another study. We also performed the reciprocal comparison. Note that the two numbers may differ if two coordinates in one study are close to one coordinate in the other; hence, we include both numbers in the Venn diagrams in *Figure 1—figure supplement 2*. To determine the extent of overlap between lists we calculated the proportion of the genome that is within the given distance threshold of each genome coordinate in a single list. We then divided this number by double the genome size (doubled to account for the two possible strands), to determine the frequency with which a randomly selected genomic position would overlap with a genome coordinate in that list. We assessed the statistical significance of the overlap between lists of genome coordinates by using a hypergeometric test to compare the number of overlapping positions and the frequency expected by chance. For example, in *Figure 1—figure supplements 2C*, 19 of the 296 3′ ends in the LB 0.4 dataset are within 10 nt of a 3′ end in the Dar and Sorek dataset. We determined how many genomic positions are within 10 nt of a position in the Dar and Sorek dataset (22899 positions). The input parameters for the hypergeometric test were 219 (number of successes), 296 (sample size), 22899 (number of successes in the population), and 9283304 (population size; twice the genome length, to account for the two possible orientations). The $p$ values are reported as $<2.2e^{-16}$ since that is the machine epsilon for 64-bit double precision values in R and Python.

## β-galactosidase assays

Rho transcriptional and MdtU translational reporter assays were performed as previously described (*Baniulyte et al., 2017*). Briefly, the pMM1constructs (*Supplementary file 4*) were assayed in MG1655Δ*lacZ* (AMD054) and MG1655Δ*lacZ rhoR66S* (GB4) backgrounds. Three separate colonies were grown overnight in LB with 30 µg/ml chloramphenicol, diluted 1:100 in the same medium, and grown to a final $OD_{600}$ ~ 0.4–0.6 at 37 ˚C. Cells were lysed in Z buffer (0.06 M $Na_2HPO_4$, 0.04 M $NaH_2PO_4$, 0.01 M KCl, 0.001 M $MgSO_4$), supplemented with β-mercaptoethanol (50 mM final concentration), sodium dodecyl sulfate (0.001% final concentration), and chloroform. Assays were initiated by adding 2-nitrophenyl β-D-galactopyranoside and stopped by adding $Na_2CO_3$. All assays were done at room temperature. The $OD_{600}$ and $A_{420}$ of the cultures were measured using a Jenway 6305 spectrophotometer. The translational *chiP-lacZ* fusions (DJS2979 and DJS2991) were assayed as above, with the following changes. Three separate colonies were grown overnight in LB with 100 µg/ml ampicillin, diluted to an $OD_{600}$ of 0.05 in the same medium supplemented with 0.2% arabinose and 1 mM IPTG, and grown for 150 min ($OD_{600}$ ~ 1.5) at 37 ˚C. Reactions were performed at 28 ˚C and the $OD_{600}$ and $A_{420}$ of the cultures were measured using an Ultrospec 3300 *pro* spectrophotometer (Amersham Biosciences). For all experiments, β-galactosidase activity was calculated as $(1000 \times A_{420})/(OD_{600} \times V_{ml} \times time_{min})$.

## Northern blot analysis

Northern blots were performed using total RNA exactly as described previously (*Melamed et al., 2020*). For small RNAs, 5 µg of RNA were fractionated on 8% polyacrylamide urea gels containing 6 M urea (1:4 mix of Ureagel Complete to Ureagel-8 (National Diagnostics) with 0.08% ammonium persulfate) and transferred to a Zeta-Probe GT membrane (Bio-Rad). For longer RNAs, 10 µg of RNA were fractionated on a 2% NuSieve 3:1 agarose (Lonza), 1X MOPS, 2% formaldehyde gel and

transferred to a Zeta-Probe GT membrane (Bio-Rad) via capillary action overnight. For both types of blots, the RNA was crosslinked to the membranes by UV irradiation. RiboRuler High Range and Low Range RNA ladders (Thermo Fisher Scientific) were marked by UV-shadowing. Membranes were blocked in ULTRAhyb-Oligo Hybridization Buffer (Ambion) and hybridized with 5′ $^{32}$P-end labeled oligonucleotides probes (listed in *Supplementary file 4*). After an overnight incubation, the membranes were rinsed with 2X SSC/0.1% SDS and 0.2X SSC/0.1% SDS prior to exposure on film. Blots were stripped by two 7 min incubations in boiling 0.2% SDS followed by two 7 min incubations in boiling water.

## Immunoblot analysis

Immunoblot analysis was performed as described previously with minor changes (*Zhang et al., 2002*). Samples were separated on a Mini-PROTEAN TGX 5–20% Tris-Glycine gel (Bio-Rad) and transferred to a nitrocellulose membrane (Thermo Fisher Scientific). Membranes were blocked in 1X TBST containing 5% milk, probed with a 1:2000 dilution of monoclonal α-FLAG-HRP (Sigma) or a 1:500 dilution of polyclonal α-OmpC follwed by a 1:1000 dilution of peroxidase labeled α-rabbit and developed with SuperSignal West Pico PLUS Chemiluminescent Substrate (Thermo Fisher Scientific) on a Bio-Rad ChemiDoc MP Imaging System.

## Genome browsers

The processed RNA-seq data from this study are available online via UCSC genome browser at the following links:

1. *E. coli* Term-seq: https://www.nichd.nih.gov/research/atNICHD/Investigators/storz/ecoli-term-seq
2. *E. coli* Rho-dependent 3′ ends (Term-seq LB 0.4 and DirectRNA-seq): https://www.nichd.nih.gov/research/atNICHD/Investigators/storz/ecoli-rho-dependent-3-ends.

# Acknowledgements

We thank S Melamed, T Updegrove, A Wagh, L Walling, K Yoo, A Zhang, and H Zhang for experimental and technical assistance, and K Papenfort, T Updegrove and members of the Storz lab for helpful comments on the manuscript. We also thank M Gottesman for the gift of bicyclomycin, the Wadsworth Center Applied Genomic Technologies and Media Core Facilities for technical assistance, and the NICHD Molecular Genomics Core, particularly T Li, for library sequencing. This work utilized the computational resources of the NIH HPC Biowulf cluster (http://hpc.nih.gov).

# Additional information

### Competing interests

Joseph T Wade: Reviewing editor, *eLife*. Gisela Storz: Senior editor, *eLife*. The other authors declare that no competing interests exist.

### Funding

| Funder | Grant reference number | Author |
| --- | --- | --- |
| National Institutes of Health | 1Fi2GM133345-01 | Philip P Adams |
| National Institutes of Health | 1ZIAHD001608-28 | Gisela Storz |
| National Institutes of Health | 1DP2OD007188 | Joseph T Wade |
| National Science Foundation | DBI 1757170 | Molly Monge |
| National Institutes of Health | 1ZICHD008986-01 | Ryan K Dale |

The funders had no role in study design, data collection and interpretation, or the decision to submit the work for publication.

## Author contributions

Philip P Adams, Conceptualization, Data curation, Formal analysis, Funding acquisition, Validation, Investigation, Visualization, Methodology, Writing - original draft, Writing - review and editing, generated the Term-seq and total RNA-seq libraries, performed all RNA-based experiments, performed the β-galactosidase assays; Gabriele Baniulyte, Data curation, Software, Formal analysis, Validation, Investigation, Visualization, Writing - review and editing, performed the β-galactosidase assays, performed the bioinformatic analyses; Caroline Esnault, Data curation, Software, Formal analysis, Visualization, Methodology, Writing - review and editing, performed the bioinformatic analyses; Kavya Chegireddy, Molly Monge, Investigation, performed the β-galactosidase assays; Navjot Singh, Investigation, prepared samples for DirectRNA-seq; Ryan K Dale, Software, Supervision, Methodology, Writing - review and editing; Gisela Storz, Joseph T Wade, Conceptualization, Formal analysis, Supervision, Funding acquisition, Visualization, Methodology, Writing - original draft, Project administration, Writing - review and editing

## Author ORCIDs

Philip P Adams [ID] https://orcid.org/0000-0002-2575-6328
Gabriele Baniulyte [ID] https://orcid.org/0000-0003-0235-7938
Gisela Storz [ID] https://orcid.org/0000-0001-6698-1241
Joseph T Wade [ID] https://orcid.org/0000-0002-9779-3160

## Decision letter and Author response

Decision letter https://doi.org/10.7554/eLife.62438.sa1
Author response https://doi.org/10.7554/eLife.62438.sa2

---

# Additional files

## Supplementary files

• Supplementary file 1. 3′ ends identified by Term-seq. Curated 3′ ends for *E. coli* MG1655 (WT) grown to $OD_{600}$ ~0.4 and 2.0 in LB, and to $OD_{600}$ ~0.4 in minimal (M63) glucose medium. The genomic coordinate of the 3′ end (3′ end position), DNA strand, average RNA-seq read count of the 3′ end of both biological replicates, 3′ end classification (see Materials and methods), the gene annotation for the classification (details), and the sequence surrounding the 3′ end (40 bp upstream and 10 bp downstream, 3′ end nucleotide red and bolded) make up the columns of the table. The data for each growth condition are displayed on a separate tab.

• Supplementary file 2. Identification of Rho termination regions using DirectRNA-seq ±BCM. All identified Rho termination regions are represented, defined as regions with at least one genomic coordinate with a significance score $<1e^{-4}$. Rho scores were calculated for each genomic position by comparing DirectRNA-seq coverage in windows 800 nt upstream and downstream in the treated (+BCM) and untreated (-BCM) samples (see materials and methods). The 3′ genomic coordinate with the highest Rho score, DNA strand, 3′ end classification (see materials and methods), the gene annotation for the classification (details), the read coverage in the 800 nt windows upstream and downstream the 3′ end position ±BCM, the Rho score, and the *p*-value from the Fisher's exact test (significance score) make up the columns of the table. An 'undefined' Rho score indicates one that could not be calculated due to zero reads in the -BCM downstream region. A significance score of 'N/A' indicates that the significance score was too low to accurately report.

• Supplementary file 3. Analysis of 3′ ends in 5′ UTRs and within coding sequences. Term-seq identified 3′ ends for *E. coli* MG1655 (WT) that were between 200 nt upstream of an ORF and the corresponding stop codon. The genomic coordinate of the 3′ end (3′ end position), DNA strand, average RNA-seq read count of the 3′ end of both biological replicates, 3′ end classification (see materials and methods), the gene annotation for the classification (details), the location of the 3′ end relative to an ORF – upstream ORF or internal (ORF classification), and the gene annotation of the associated ORF (upstream ORF/internal details) make up the columns of the table. The data for each growth condition are displayed on a separate tab. For 3′ ends identified in WT *E. coli* grown to $OD_{600}$ ~0.4 in LB, each 3′ end was also given a Rho score, an assessment of whether it is in a Rho

termination region, and an intrinsic terminator score (see materials and methods). The read coverage in 800 nt windows upstream and downstream of each 3′ end position ±BCM from the DirectRNA-seq data, used to calculate Rho scores, is presented. An 'undefined' Rho score indicates one that could not be calculated due to zero reads in the ±BCM downstream region. Intrinsic terminator scores > 3.0 are suggestive of intrinsic termination (*Chen et al., 2013*). An 'undefined' intrinsic terminator score indicates one that could not be calculated because the sequence could not be folded into a recognizable secondary structure. Characterized regulatory elements in the LB 0.4 condition are also noted.

• Supplementary file 4. List of strains together with plasmids (tab 1) and oligonucleotides (tab 2) used in this study.

• Transparent reporting form

### Data availability

The raw sequencing data reported in this paper have been deposited in SRA under accession number PRJNA640168. Code for calling 3′ ends in Term-seq sequencing reads can be found at https://github.com/NICHD-BSPC/termseq-peaks (copy archived at https://archive.softwareheritage.org/swh:1:rev:8bb48d2a034b22a312a7e848e0f8694a284a8324/). Code for calling Rho termination regions can be found at https://github.com/gbaniulyte/rhoterm-peaks (copy archived at https://archive.softwareheritage.org/swh:1:rev:831f7cb906d0af5767ffbfbb63ec247579d5e250/). The processed RNA-seq data from this study are available online via UCSC genome browser at the following links: E. coli Term-seq: https://www.nichd.nih.gov/research/atNICHD/Investigators/storz/ecoli-term-seq E. coli Rho-dependent 3′ ends (Term-seq LB 0.4 and DirectRNA-seq): https://www.nichd.nih.gov/research/atNICHD/Investigators/storz/ecoli-rho-dependent-3-ends.

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
