## [Decision Letter]

**Acceptance summary:**

This study expands our definition of bacterial small RNAs (sRNAs) as it demonstrates functionality of several "nonconventional" sRNAs. The work is expected to boost future studies looking into bacterial sRNAs derived from 5'UTRs or ORFs in *E. coli* and beyond.

**Decision letter after peer review:**

Thank you for submitting your article "Regulatory roles of 5' UTR and ORF-internal RNAs detected by 3' end mapping" for consideration by *eLife*. Your article has been reviewed by four peer reviewers, including Alexander Westermann as the Reviewing Editor and Reviewer #1, and the evaluation has been overseen by Kevin Struhl as the Senior Editor. The following individual involved in review of your submission has agreed to reveal their identity: Masatoshi Miyakoshi (Reviewer #2).

The reviewers have discussed the reviews with one another and the Reviewing Editor has drafted this decision to help you prepare a revised submission.

Summary:

In the present study, you have comprehensively identified the 3' ends of transcripts in *E. coli* and demonstrated that many arise from premature transcription termination in either Rho-dependent or intrinsic manner. As a result, you discovered numerous stable RNAs derived from 5'UTRs or CDSs and functionally characterized several of these "unconventional" RNAs as sponges of well-studied Hfq-dependent small RNAs. The reviewers all agreed that this is impressive work, the findings are novel and relevant for researchers within the microbiology and RNA communities and may inspire future studies of non-canonical bacterial sRNAs. Overall, they deem the results convincingly supported by the experimental data, but would like to see a few more experimental and analytical amendments to your work.

Essential revisions:

1) Comparison of global data sets: For cross-comparison, it would be advisable that the current data sets and previously published ones were analyzed consistently. This might increase the overlap between the results of the different studies.

a) Term-seq: The computational method used to process the current Term-seq data is different from the one presented in the original Term-seq paper of Dar and Sorek. The authors should explain why they turned to a different computational pipeline and – for cross-comparison – reanalyze the published data set from Dar and Sorek with their own computational methodology, or analyze their results by Dar and Sorek's computational method.

b) Rho-dependent termination: Since the authors here made the effort to treat the cells with BCM and generate sequencing libraries, it is not clear why they did not simply carry out Term-seq following BCM treatment and compared the identified 3' ends to those determined without BCM. Rather, the authors followed the analysis pipeline of Dar and Sorek who used available data of BCM-treated cells from Peters et al., 2012, and therefore could only evaluate the readthrough in the vicinity of determined 3' ends. Here also, the authors modified the computational method of Dar and Sorek. This needs justification and the parameters used should be explained (e.g. why using a window size of 500 nt and threshold of the Rho score of 2). For cross-comparison of the results, the Dar and Sorek data set and the current data set should be analyzed by the same method.

2) Evaluation of statistical significance: It is not always clear how the reported p-values of the hypergeometric tests were computed, and it is not possible to re-compute them as the value of N was not provided. Please verify that p (x{greater than or equal to} actual result) was computed and provide the details of the computation for all hypergeometric tests included in the manuscript.

3) Experimental validation: Several newly discovered 3' termination sites were tested experimentally. From the reported results it seems that all tested sites were validated by wet-lab experiments. Could the authors explain how the individual examples were selected? Were there any predicted 3' ends that could not be validated experimentally? If so, reporting true vs. false positives would provide another assessment of the reliability of the data.

4) Rho-dependent premature termination (–subsection “Novel sites of regulation are predicted by 3´ ends and Rho termination regions in 5´ UTRs”; Figure 2D, E, F): The results obtained from some of the northern blots seem confusing in light of the corresponding LacZ reporter assays. For example, the galactosidase experiments with cells from OD 0.4~0.6 showed an increase of LacZ activity in the rhoR66S mutant for most reporters, as expected (Figure 2D, E and Figure 2—figure supplement 1B). On the other hand, in several cases the northern blot analysis of total RNA extracted from cells at OD 0.4 revealed the increase of prematurely terminated 5'UTR fragments in the rhoR66S strain (Figure 2F and Figure 2—figure supplement 1C). Wouldn't these 5' fragments be expected to accumulate in the WT – rather than the R66S – strain, if they were generated through Rho-dependent termination? The authors' hypothesis that increased levels of longer transcripts in the absence of Rho could be processed to give rise to these shorter products (see the aforementioned subsection) could be better explained. In parallel, the northern blot membranes should be re-hybridized with probes for some of the sequences downstream of the termination sites to corroborate this assumption.

5) sRNA sponges: The exact effect of RybB on FtsO has not been clarified in the manuscript (subsection “ORF-internal FtsO is an sRNA sponge”). When RybB is abundant, the level of FtsO is reduced (Figure 7B). This may be indicative of coupled degradation upon base-pairing between FtsO and RybB. However, when RybB was induced by ethanol (Figure 7E), the level of FtsO was unchanged (or even increased), probably attributable to transcriptional activation of ftsI. To clarify the reciprocal regulation between RybB and FtsO and the consequences of their interaction, the half-life of each sRNA in the presence or absence of its counterpart sRNA should be quantified. Additionally, the (indirect) effect of FtsO on the RybB target *ompC* is not very pronounced (Figure 7E). Could the authors please quantify this effect on the protein level (e.g. by western blot, ß-gal assay) to further support the notion that FtsO-mediated sponging of RybB translates into a de-repression of RybB targets?

---

## [Author Response]

Essential revisions:1) Comparison of global data sets: For cross-comparison, it would be advisable that the current data sets and previously published ones were analyzed consistently. This might increase the overlap between the results of the different studies.

We encountered a number of complications that prevented uniform comparisons. For example, the Term-seq paper of Dar and Sorek does not globally identify 3´ ends and rather only reports 3´ ends directly downstream of annotated ORFs. Therefore, we only compared our “primary 3´ ends” (downstream of annotated ORFs) to this dataset. As another example, the study by Ju et al. published one table of 3´ ends identified from logarithmic and stationary phase grown bacteria together, without distinguishing 3´ ends identified in each condition. Therefore, we compared our 3´ ends identified from LB 0.4 and 2.0 conditions to this dataset. We feel that this approach most accurately compares the data.

a) Term-seq: The computational method used to process the current Term-seq data is different from the one presented in the original Term-seq paper of Dar and Sorek. The authors should explain why they turned to a different computational pipeline and – for cross-comparison – reanalyze the published data set from Dar and Sorek with their own computational methodology, or analyze their results by Dar and Sorek's computational method.

Briefly, the method reported by Dar and Sorek was based on retaining the most-highly covered, reproducibly-occurring 3´ end downstream of gene such that a kept position had to be detected in at least 2 independent experiments with an average coverage of at least 4 reads across all three replicates. The “at least 4 reads” threshold of the Dar and Sorek method is completely dependent on the sequencing depth of the particular replicate and the characteristics of the experiment, so it is difficult to generalize this threshold to other experiments. Furthermore the Dar and Sorek method does not take into account replicate variability.

Our method instead starts with a peak-calling step using the scipy.signal.find_peaks function commonly used in electronic signal processing. This is then followed by a statistically-founded comparison of replicate reproducibility (irreproducible discovery rate, IDR) which uses a rank-based approach to identify peaks that are both strong and consistent among replicates. Taken together, these two steps, as implemented in our open-source term-seq peak-calling package https://github.com/NICHD-BSPC/termseq-peaks result in a method that is agnostic to sample coverage and that at the same time takes into account replicate variability by scoring highly those peaks found at consistent ranks across replicates. Empirically, we observe that our method´s results match manual observations in the genome browser much better than the Dar and Sorek approach. Furthermore, we have successfully used the same method with two other bacterial species (in unpublished work), supporting the generality of our approach.

As suggested, we reanalyzed the data in the Dar and Sorek paper using our algorithm and compared these 3´ ends to ours. The results have been added to Figure 1—figure supplement 2. The following statement has been added to the main text “Given that a previous *E. coli* Term-seq study only reported 3´ ends downstream annotated genes (Dar and Sorek, 2018b), we also used our 3´ end calling algorithm to re-analyze the sequencing data from this study (Figure 1—figure supplement 2D). Again, there was significant overlap (hypergeometric test *p* < 2.2e^-16^), but 51% were unique to our study.”

b) Rho-dependent termination: Since the authors here made the effort to treat the cells with BCM and generate sequencing libraries, it is not clear why they did not simply carry out Term-seq following BCM treatment and compared the identified 3' ends to those determined without BCM. Rather, the authors followed the analysis pipeline of Dar and Sorek who used available data of BCM-treated cells from Peters et al., 2012, and therefore could only evaluate the readthrough in the vicinity of determined 3' ends. Here also, the authors modified the computational method of Dar and Sorek. This needs justification and the parameters used should be explained (e.g. why using a window size of 500 nt and threshold of the Rho score of 2). For cross-comparison of the results, the Dar and Sorek data set and the current data set should be analyzed by the same method.

Term-seq +/-BCM would not be effective for identification of all Rho-dependent terminators since our data show that, in many cases, 3´ end abundance does not decrease following BCM treatment. This is presumably because the 3´ ends are generated by RNase processing rather than directly by Rho, and the processing can occur even when transcription reads through the terminator. Our approach assessed the Rho scores for all regions in the DirectRNA-seq data (not only for 3´ ends identified by Term-seq, although we did score each of the Term-seq 3´ ends specifically for Rho termination). Supplementary file 2 is a list of all the putative Rho termination regions identified from DirectRNA-seq alone.

The window size and thresholds we used were somewhat arbitrary. Prompted by the reviewer comments, we performed an empirical analysis of these parameters, allowing us to optimize the sensitivity and specificity of our approach. As a consequence, we made several improvements to our analysis pipeline, although the basic approach is unchanged: we count the number of sequence reads in DirectRNA-seq data in a window upstream and a window downstream of the genome coordinate of interest, for untreated and BCM-treated datasets. We then compare these numbers, looking for cases where the downstream:upstream ratio is higher in the BCM-treated dataset than in the untreated dataset. While the new pipeline is fundamentally similar to the old one, there are several important improvements. First, we no longer use the Rho score to set a cut-off for calling Rho-dependent terminators. Using the Rho score required us to set a threshold for the minimum number of reads in each window, to avoid taking ratios of very small numbers. We now use a Fisher’s exact test to compare the scores for each window, obviating the need for a minimum read count threshold.

Second, we have increased the window size to 800 nt. This is based on a thorough ROC analysis (see Author response image 1). Specifically, we selected the 50 most Rho-dependent, and the 50 least Rho-dependent terminators identified by Dar and Sorek; these served as true positive and true negatives, respectively. We then calculated the true positive rate (TPR) and false positive rate (FPR) using our method with different p-value cut-offs (From the Fisher’s exact test). Thus, we generated ROC plots that show FPR and TPR across a range of p-value cut-offs (see Author response image 1, left panel, for an 800 nt window size). We generated ROC plots for different window sizes (100 -1000 nt, in 100 nt increments). The window size that gave the highest area under the curve in the ROC plots was 800 nt (see Author response image 1, right panel). We also carried out similar analysis using different upstream and downstream window sizes, including the window sizes used by Dar and Sorek, but the ROC area under the curve values were invariably lower than those we determined for an 800 nt window on both sides. Using an 800 nt window size, we selected the p-value cut-off that gives the largest value of TPR-FPR, which in this case is a difference of 0.64 (TPR = 0.72, FPR = 0.08; circled datapoint in Author response image 1, left panel). Finally, we applied these parameters to call peaks from the DirectRNA-seq data, identifying 1,078 putative sites of Rho-dependent termination. Importantly, the regions around these sites have a similar C:G ratio to the previous list (Figure 1D), and to putative sites of Rho termination identified by Peters et al.

2) Evaluation of statistical significance: It is not always clear how the reported p-values of the hypergeometric tests were computed, and it is not possible to re-compute them as the value of N was not provided. Please verify that p (x ≥ actual result) was computed and provide the details of the computation for all hypergeometric tests included in the manuscript.

A description of the hypergeometric tests we carried out was added to the Materials and methods section. The numbers used for the tests were taken from each of the individual figures. Note that the statistical significance of the overlaps is extremely high because the number of genomic coordinates analyzed is high, however, the *p* values are reported as < 2.2e^-16^ since that is the machine epsilon for 64-bit double precision values in R and Python.

The following example was added to the text to allow readers to re-compute the p values. “For example, in Figure 1—figure supplement 2C, 219 of the 296 3´ ends in the LB 0.4 dataset are within 10 nt of a 3´ end in the Dar and Sorek dataset. We determined how many genomic positions are within 10 nt of a position in the Dar and Sorek dataset (22899 positions). The input parameters for the hypergeometric test were 219 (number of successes), 296 (sample size), 22899 (number of successes in the population), and 9283304 (population size; twice the genome length, to account for the two possible orientations).”

3) Experimental validation: Several newly discovered 3' termination sites were tested experimentally. From the reported results it seems that all tested sites were validated by wet-lab experiments. Could the authors explain how the individual examples were selected? Were there any predicted 3' ends that could not be validated experimentally? If so, reporting true vs. false positives would provide another assessment of the reliability of the data.

We selected sites with a cross-section of range of Rho scores for the 28 examples assayed by the *lacZ* fusions. All 3´ ends that we tested are shown. Based on the fold induction compared to the Rho scores, five of these examples could be false positives for Rho-mediated regulation. The examples selected for the northern analysis were chosen because of other known or predicted regulation. We think the numbers assayed by northern analysis are too few to be able to comment on true versus false positives. The text was changed to better convey this information.

4) Rho-dependent premature termination (–subsection “Novel sites of regulation are predicted by 3´ ends and Rho termination regions in 5´ UTRs”; Figure 2D, E, F): The results obtained from some of the northern blots seem confusing in light of the corresponding LacZ reporter assays. For example, the galactosidase experiments with cells from OD 0.4~0.6 showed an increase of LacZ activity in the rhoR66S mutant for most reporters, as expected (Figure 2D, E and Figure 2—figure supplement 1B). On the other hand, in several cases the northern blot analysis of total RNA extracted from cells at OD 0.4 revealed the increase of prematurely terminated 5'UTR fragments in the rhoR66S strain (Figure 2F and Figure 2—figure supplement 1C). Wouldn't these 5' fragments be expected to accumulate in the WT – rather than the R66S – strain, if they were generated through Rho-dependent termination? The authors' hypothesis that increased levels of longer transcripts in the absence of Rho could be processed to give rise to these shorter products (see the aforementioned subsection) could be better explained. In parallel, the northern blot membranes should be re-hybridized with probes for some of the sequences downstream of the termination sites to corroborate this assumption.

We were also very surprised by this accumulation of 5´ UTR fragments in the rhoR66S strain but found this to be true in many of the examples we examined. The 3´ ends generated following premature Rho termination are likely due to processing by exonucleases. This can still happen in transcripts where RNA polymerase has read through the terminator, assuming there is an endonuclease that cuts downstream.

As suggested by the reviewers, we now try to explain our hypothesis better. We also used the RNA samples in Figure 2F for an agarose gel northern, which we probed for the coding sequences (downstream of the identified 3´ ends) for *cfa*, *cyaA*, *speA* and *mdtJI*. These blots, added to Figure 2—figure supplement 1, showed much higher levels of read-through in the rhoR66S mutant.

5) sRNA sponges: The exact effect of RybB on FtsO has not been clarified in the manuscript (subsection “ORF-internal FtsO is an sRNA sponge”). When RybB is abundant, the level of FtsO is reduced (Figure 7B). This may be indicative of coupled degradation upon base-pairing between FtsO and RybB. However, when RybB was induced by ethanol (Figure 7E), the level of FtsO was unchanged (or even increased), probably attributable to transcriptional activation of ftsI. To clarify the reciprocal regulation between RybB and FtsO and the consequences of their interaction, the half-life of each sRNA in the presence or absence of its counterpart sRNA should be quantified. Additionally, the (indirect) effect of FtsO on the RybB target ompC is not very pronounced (Figure 7E). Could the authors please quantify this effect on the protein level (e.g. by western blot, ß-gal assay) to further support the notion that FtsO-mediated sponging of RybB translates into a de-repression of RybB targets?

We have carried out a number of additional experiments to address the several different issues raised in this comment. (i) To test whether there is coupled degradation upon base pairing between FtsO and RybB, we examined the effect of RybB overexpression on FtsO levels (Figure 8—figure supplement 1A). We found that indeed, there is a reciprocal effect of RybB on FtsO for stationary phase cells. This is now discussed in the text. (ii) To further test the effect of FtsO on the RybB target *ompC*, we compared *ompC* mRNA and protein levels upon overexpression of FtsO and the previously characterized RybB sponge, RbsZ (Figure 8—figure supplement 1B). With both FtsO and RbsZ we observed clear increases in the levels of the *ompC* mRNA, but no obvious effects on the OmpC protein. This is likely due to abundant, stable levels of OmpC present at this condition. This is now stated and discussed in the text. We also re-ran the agarose northern in Figure 8E (previously Figure 7E) to have a better quality 5S signal.